# Planning in Stochastic Environments with a Learned Model

**Ioannis Antonoglou**[1,2]    **Julian Schrittwieser**[1]    **Sherjil Ozair**[1]    **Thomas Hubert**[1]

**David Silver**[1,2]

[1]DeepMind, London, UK
[2]University College London

## Abstract

Model-based reinforcement learning has proven highly successful. However, learning a model in isolation from its use during planning is problematic in complex environments. To date, the most effective techniques have instead combined value-equivalent model learning with powerful tree-search methods. This approach is exemplified by *MuZero*, which has achieved state-of-the-art performance in a wide range of domains, from board games to visually rich environments, with discrete and continuous action spaces, in online and offline settings. However, previous instantiations of this approach were limited to the use of deterministic models. This limits their performance in environments that are inherently stochastic, partially observed, or so large and complex that they appear stochastic to a finite agent. In this paper we extend this approach to learn and plan with stochastic models. Specifically, we introduce a new algorithm, *Stochastic MuZero*, that learns a stochastic model incorporating afterstates, and uses this model to perform a stochastic tree search. *Stochastic MuZero* matched or exceeded the state of the art in a set of canonical single and multi-agent environments, including *2048* and backgammon, while maintaining the superhuman performance of standard *MuZero* in the game of Go.

## 1 Introduction

Constructing plans and executing them is an important feature of human and animal behaviour. In the field of artificial intelligence there has been a great amount of research into adding planning capabilities to intelligent agents. Tree-based planning algorithms have shown a lot of success in a wide variety of environments such as card games (Moravčík et al., 2017), board games (Campbell et al., 2002; Silver et al., 2016) and more recently video games (Schrittwieser et al., 2020) and continuous control tasks (Hubert et al., 2021). Most tree search methods assume that the agent has access to a perfect simulator of the environment, whereas real-world environments are typically unknown.

Model-based reinforcement learning algorithms combine a model-learning component, which estimates the dynamics of the environment, with a planning component, using the learned model as a simulator. However, learning a model in isolation from its use during planning has proven to be problematic in complex environments (van Hasselt et al., 2019). Instead, value-equivalent model-learning methods (Silver et al., 2017; Farahmand et al., 2017; Oh et al., 2017; Grimm et al., 2020) identify a model that reconstructs only those quantities required for planning. The most successful method, *MuZero* (Schrittwieser et al., 2020) learns a model that reconstructs reward, value and policy, and uses this model to perform a powerful Monte Carlo tree search. *MuZero* achieved superhuman results in Go, chess, shogi and Atari without any prior knowledge of the rules, and has also achieved state-of-the-art performance in large and continuous action spaces (Hubert et al., 2021) and offline reinforcement learning (Schrittwieser et al., 2021).

However, value equivalent methods such as *MuZero* have in practice been limited to a deterministic class of models, which severely limits their applicability. Many environments are inherently stochastic

and may be poorly approximated by a deterministic model. Partially observed environments may also be perceived by the agent as stochastic, whenever aliased states cannot be disambiguated. Similarly, large and complex environments may appear stochastic to a small agent with finite capacity.

In this paper we introduce the first empirically effective approach for handling stochasticity in value equivalent model-learning and planning. The model is factored to first transition deterministically from state to an afterstate, and then to branch stochastically from the afterstate to the next state. This factored model is trained end-to-end so as to maintain value equivalence for both state value function and action value function respectively, and a stochastic planning method is applied to the model.

We apply these ideas to *MuZero*, using a discrete generative network to represent the model, and modifying the Monte Carlo tree search to effectively use the factored model. We apply our method, *Stochastic MuZero*, to several environments in which handling stochasticity is important. First, we consider the popular stochastic puzzle game *2048*, in which the prior state of the art exploits a perfect simulator and significant handcrafted domain knowledge. In our experiments, *Stochastic MuZero* achieved better results without any domain knowledge. Second, we consider the classic stochastic two-player game of backgammon, in which near-optimal play has been achieved using a perfect simulator. *Stochastic MuZero* matches this performance without any prior knowledge of the game rules. Finally, we evaluated our method in the deterministic board game of Go. There our method matched the performance of *MuZero*, demonstrating that *Stochastic MuZero* extends *MuZero* without sacrificing performance.

## 2 RELATED WORK

**Observation models**   (Oh et al., 2015; Chiappa et al., 2017; Łukasz Kaiser et al., 2020) explicitly learn the dynamics of an environment by fitting a model of observations and rewards to observed transitions. Subsequently, these models can be combined with a model-free learning rule in a Dyna fashion (Sutton, 1991). However, modeling high dimensional image observations can be computationally prohibitive, prone to high error accumulation as the model is unrolled for multiple steps, and limiting since the capacity of the model could be spent on background features which are not helpful for the problem at hand. These issues make such models unconducive for planning. Finally, van Hasselt et al. (2019) argues that Dyna-based methods are unlikely to outperform model-free approaches that use a replay buffer.

**Latent models**   (Schrittwieser et al., 2020; Oh et al., 2017; Hafner et al., 2021; Henaff et al., 2017) attempt to overcome the limitations of observation models by learning recurrent networks that operate on latent states. In this framework, the model is conditioned on the current observation and future actions and is unrolled for $k$ steps. Subsequently, it is trained to make predictions about rewards, values, policies or observations at each timestep based on the current latent state. The model can then be combined with a tree-based planning algorithm or used to generate synthetic trajectories. Recently, *MuZero* has shown that it is possible to use this approach to achieve state-of-the-art performance in many challenging domains (Hubert et al., 2021) while using less data (Schrittwieser et al., 2021). However, most approaches, including *MuZero*, use a deterministic function to model the environment dynamics, which limits their applicability to deterministic or weakly stochastic environments.

**Stochastic latent models**   are stochastic models of the environment dynamics that operate on latent states. In (Hafner et al., 2021) the authors propose a recurrent state-space model which consists of three main modules, a recurrent module which generates the deterministic recurrent state $h_t$, a representation model which combines $h_t$ with the current observation $x_t$ to generate a distribution over stochastic states $s_t$ and plays the role of the posterior, and a transition predictor which depends only on $h_t$ and acts as the prior of the model. By combining the deterministic and stochastic states $h_t$ and $s_t$ the model is trained to predict the current observation $o_t$, the transition reward $r_t$ and the discount $d_t$. The next deterministic recurrent state is generated using $h_t$, $s_t$ and action $a_t$. The stochastic states $s_t$ are modeled as multidimensional multinomial variables. The learned model is then used to generate synthetic data which are used to train an actor-critic model-free agent. The authors show that their approach outperforms pure model-free methods but it fails to achieve the performance of *MuZero* which combines its learned model with planning.

In (Ozair et al., 2021) the authors learn a stochastic transition model using a VQ-VAE generative network (van den Oord et al., 2017) and subsequently combine it with MCTS. They show that their method can match the performance of *MuZero* in chess, while viewing the problem as a single-player task and implicitly learning to model the behaviour of the opponent. Despite its promise their approach was only applied in a supervised setting using expert data, and did not address the challenges of learning a stochastic model in the reinforcement learning setting. Moreover, the learned model was trained to explicitly predict the observation at every step, which can be a limiting factor in terms of computation and model efficiency when dealing with high dimensional observations. Finally, the authors used a two stage training process: first, a model learns latent representations of the observations, then these representations are used to learn a transition model. This makes it hard to apply this approach in the reinforcement learning setting.

## 3 BACKGROUND

### 3.1 *MuZero*

*MuZero* is a model-based general reinforcement learning agent which combines a learned model of the environment dynamics with a *Monte Carlo tree search* planning algorithm. The model is conditioned on the history of observations $o_{\leq t}$ at timestep $t$ and a sequence of future actions $a_{t:t+K}$, and it is trained to predict the search policies $\pi_{t:t+K}$, values $v^{\pi}_{t:t+K}$ and intermediate rewards $r_{t:t+K}$ at each future timestep. *MuZero* uses deterministic functions for its model, and thus it implicitly assumes that the underlying environment dynamics are also deterministic. *MuZero* uses its dynamics model to plan ahead at each time step and the outcome of its MCTS search to select an action and as targets for its policy improvement operator.

**Model** *MuZero*'s learned model consists of 3 functions: a *representation* function $h$, a dynamics function $g$ and a prediction function $f$. The *representation* function maps the current history of observations $o_{\leq t}$ into a latent state $s^0_t$. The *dynamics* function $g$ receives the previous latent state $s^k_t$ and combines it with an action $a_{t+k}$ to produce the next latent state $s^{k+1}_t$ and the reward $r^k_t$. Finally, the *prediction* function $f$ receives each latent state $s^k_t$ as an input and computes the policy $p^k_t$ and value $v^k_t$. Given a sequence of policy $\pi_{t:T}$, value $z_{t:T}$, and reward $u_{t:T}$ targets, the model is trained to minimize the loss shown in 1.

$$L^{MuZero} = \sum_{k=0}^{K} l^p(\pi_{t+k}, p^k_t) + \sum_{k=0}^{K} l^v(z_{t+k}, v^k_t) + \sum_{k=1}^{K} l^r(u_{t+k}, r^k_t) \qquad (1)$$

The policy targets $\pi_{t+k}$ correspond to the MCTS policy that was generated when searching from observation $o_{\leq t+k}$. The value targets $z_{t+k}$ are computed using *n-step returns* (Sutton & Barto, 2018). Finally, the reward targets $u_{t+k}$ correspond to the real instantaneous rewards observed when this sequence was generated.

**Search** *MuZero* uses a variant of the MCTS tree based algorithm first proposed in (Silver et al., 2018). The tree is constructed recursively through a number of simulations. Each simulation consists of 3 phases: *selection*, *expansion* and *backpropagation*. During the *selection* phase the tree is traversed starting from the root node until a leaf edge is reached. At each internal node $s$ the algorithm selects the action $a$ which maximizes the upper confidence bound proposed in (Silver et al., 2016) and shown in equation 2.

$$a = \arg\max_a \left[ Q(s,a) + P(a \mid s) \cdot \frac{\sqrt{1 + \sum_b N(s,b)}}{1 + N(s,a)} \left( \alpha_1 + \log\left( \frac{\sum_b N(s,b) + \alpha_2 + 1}{\alpha_2} \right) \right) \right] \quad (2)$$

Here, $Q(s,a)$ is the value estimate for action $a$, $N(s,a)$ the visit count, $P(a \mid s)$ the prior probability of selecting action $a$, and $\alpha_1$, $\alpha_2$ are constants which control the relative importance of the $Q(s,\cdot)$ estimates and prior probabilities $P(\cdot \mid s)$. In the next phase *expansion*, the leaf edge is expanded by querying the *MuZero* model and a new node is added to the tree. Finally, during the *backpropagation* phase the value estimate of the newly added edge is backpropagated up the tree using the n-step return estimate.

## 3.2 Vector Quantised Variational AutoEncoder

Vector Quantised Variational AutoEncoder (VQ-VAE, van den Oord et al. (2017)) is a generative modeling technique which uses four key components: an encoder neural network $e$, a decoder neural network $d$, a vector quantisation layer $vq$, and an autoregressive model $m$. Given an input $x_t$, the encoder produces an embedding $c_t^e = e(x_t)$. The quantisation layer comprises of a set of $M$ codes $\{c_i\}_{i=0}^M$, called the *codebook*, and quantises the encoder's output embedding $c_t^e$ by returning the nearest code $c_t = c_{k_t}$ along with its index $k_t = \arg\min_i \|c_i - c_t^e\|$. Additionally, in the backwards pass, this quantisation is treated as an identity function, referred to as *straight-through* gradient estimation (Bengio et al., 2013). The decoder produces a reconstruction of the input $\hat{x}_t = d(c_t)$. The autoregressive model predicts a distribution $p(k_t|c_{<t}) = m(c_{<t})$ over the code index at time $t$ using the quantised embeddings $c_{<t}$ of the previous timesteps. The VQ-VAE equations are shown in Equations 3.

$$
\begin{aligned}
\textit{Encoder} && c_t^e &= e(x_t) \\
\textit{Quantisation} && c_t, k_t &= vq(c_t^e) \\
\textit{Decoder} && \hat{x}_t &= d(c_t) \\
\textit{Model} && p(k_t|c_{<t}) &= m(c_{<t})
\end{aligned}
\tag{3}
$$

Typically, the encoder, decoder, and codebook are trained first and then frozen to train the autoregressive model in an additional second stage. The total loss for the VQ-VAE is

$$
L_\phi^{vqvae} = \sum_{t=0}^{N-1} \left[ \underbrace{\|\hat{x}_t - x_t\|}_{\text{reconstruction}} + \beta \underbrace{\|c_t - c_t^e\|^2}_{\text{commitment}} - \gamma \underbrace{\log p(k_t|c_{<t})}_{\text{second stage}} \right]
\tag{4}
$$

## 4  Stochastic MuZero

In this section we present our novel algorithm *Stochastic MuZero*. Our approach combines a learned stochastic transition model of the environment dynamics with a variant of Monte Carlo tree search (MCTS). First, we describe the new model and subsequently how it is combined with MCTS for planning.

### 4.1 Stochastic Model

***Afterstates***   We consider the problem of modeling the dynamics of a stochastic environment. Similarly to *MuZero*, the model receives an initial observation $o_{\leq t}$ at time step $t$ and a sequence of actions $a_{t:t+K}$, and needs to make predictions about the future values, policies and rewards. In contrast to *MuZero* which only considers latent states which correspond to real states of the environment, *Stochastic MuZero* makes use of the notion of afterstates (Sutton & Barto, 2018) to capture the stochastic dynamics. An afterstate $as_t$ is the hypothetical state of the environment after an action is applied but before the environment has transitioned to a true state:

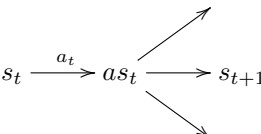

By using afterstates we can separate the effect of applying an action to the environment and of the chance transition given an action. For example in backgammon, the afterstate corresponds to the board state after one player has played its action but before the other player had the chance to roll the dice. It is also possible to define the value of an afterstate as $V(as_t) = Q(s_t, a_t)$ and the transition probabilities of the environment dynamics $Pr(s_{t+1} \mid as_t) = Pr(s_{t+1} \mid s_t, a_t)$. An afterstate can lead to multiple states based on a chance event. In our work we assume that there is a finite number of possible states $M$ that the environment can transition to, given an afterstate, and this way we can associate each transition with a chance outcome $c_t^i$. An example of a chance outcome could be the result of the dice in a game of backgammon. By defining afterstates $as_t$ and chance outcomes

$c_t$, we can model a chance transition using a deterministic model $s_{t+1}, r_{t+1} = \mathcal{M}(as_t, c_t)$ and a distribution $Pr(s_{t+1} \mid as_t) = Pr(c_t \mid as_t)$. The task of learning a stochastic model is then reduced to the problem of learning afterstates $as$ and chance outcomes $c$.

**Model** The stochastic model of *Stochastic MuZero* consists of 5 functions: a *representation* function $h$ which maps the current observation $o_{\leq t}$ to a latent state $s_t^0$, an *afterstate dynamics* function $\phi$ which given a state $s_t^k$ and an action $a_{t+k}$ produces the next latent afterstate $as_t^k$, a *dynamics* function $g$ which given an afterstate $as_t^k$ and a chance outcome $c_{t+k+1}$ produces the next latent state $s_t^{k+1}$ and a reward prediction $r_t^{k+1}$, a *prediction* function $f$ which given a state $s_t^k$ generates the value $v_t^k$ and policy $p_t^k$ predictions, and a *afterstate prediction* function $\psi$ which given an afterstate $as^k$ generates a value prediction $Q_t^k$, and a distribution $\sigma_t^k = Pr(c_{t+k+1} \mid as_t^k)$ over possible future chance outcomes $c_{t+k+1}$. The model equations are shown in 5.

$$
\begin{aligned}
\textit{Representation} && s_t^0 &= h(o_{\leq t}) \\
\textit{Prediction} && p_t^k, v_t^k &= f(s_t^k) \\
\textit{Afterstate Dynamics} && as_t^k &= \phi(s_t^k, a_{t+k}) \\
\textit{Afterstate Prediction} && \sigma_t^k, Q_t^k &= \psi(as_t^k) \\
\textit{Dynamics} && s_t^{k+1}, r_t^{k+1} &= g(as_t^k, c_{t+k+1})
\end{aligned}
\tag{5}
$$

During inference, given an initial observation $o_{\leq t}$ and a sequence of actions $a_{t:t+K}$, we can generate trajectories from the above model by recurrently unrolling it and by sampling chance outcomes from the distributions $c_{t+k+1} \sim \sigma_t^k$.

**Chance outcomes** *Stochastic MuZero* models the chance outcomes by using a novel variant of the VQ-VAE method. Specifically, we consider a VQ-VAE with a constant codebook of size $M$. Each entry in the codebook is a fixed one-hot vector of size $M$. By using a fixed codebook of one hot vectors, we can simplify the equations of the VQ-VAE 3. In this case, we model the encoder embedding $c_t^e$ as a categorical variable, and selecting the closest code $c_t$ is equivalent to computing the expression $\mathrm{one\,hot}(\arg\max_i(c_t^{e,i}))$. The resulting encoder can also be viewed as a stochastic function of the observation which makes use of the Gumbel softmax reparameterization trick (Jang et al., 2016) with zero temperature during the forward pass and a straight through estimator during the backward. There is no explicit decoder in our model, and contrary to previous work (Ozair et al., 2021) we do not make use of a reconstruction loss. Instead the network is trained end-to-end in a fashion similar to *MuZero*. In the following section we explain the training procedure in more detail.

**Model training** The stochastic model is unrolled and trained in an end-to-end fashion similar to *MuZero*. Specifically, given a trajectory of length $K$ with observations $o_{\leq t:t+K}$, actions $a_{t:t+K}$, value targets $z_{t:t+K}$, policy targets $\pi_{t:t+K}$ and rewards $u_{t+1:t+K}$, the model is unrolled for $K$ steps as shown in figure 1 and is trained to optimize the sum of two losses as shown in equation 6: a *MuZero* loss and a *chance* loss for learning the stochastic dynamics of the model.

$$
L^{total} = L^{MuZero} + L^{chance}
\tag{6}
$$

The *MuZero* loss is the same as the one described in *MuZero* (see equation 3.1). The chance loss is applied to the predictions $Q_t^k$ and $\sigma_t^k$ which correspond to the latent *afterstates* $as^k$. The $Q_t^k$ value is trained to match the value target $z_{t+k}$ and the $\sigma^k$ is trained towards the one hot chance code $c_{t+k+1} = \mathrm{one\,hot}(\arg\max_i(e(o_{\leq t+k+1}^i)))$ produced by the encoder. Finally, following the standard VQ-VAE practice, we use a VQ-VAE commitment cost to ensure that the output of the encoder $c_{t+k}^e = e(o_{\leq t+k+1})$ is close to the code $c_{t+k}$. Equation 7 shows the chance loss used to train the model.

$$
L_w^{chance} = \sum_{k=0}^{K-1} l^Q(z_{t+k}, Q_t^k) + \sum_{k=0}^{K-1} l^\sigma(c_{t+k+1}, \sigma_t^k) + \beta \underbrace{\sum_{k=0}^{K-1} \left\| c_{t+k+1} - c_{t+k+1}^e \right\|^2}_{\text{VQ-VAE commitment cost}}
\tag{7}
$$

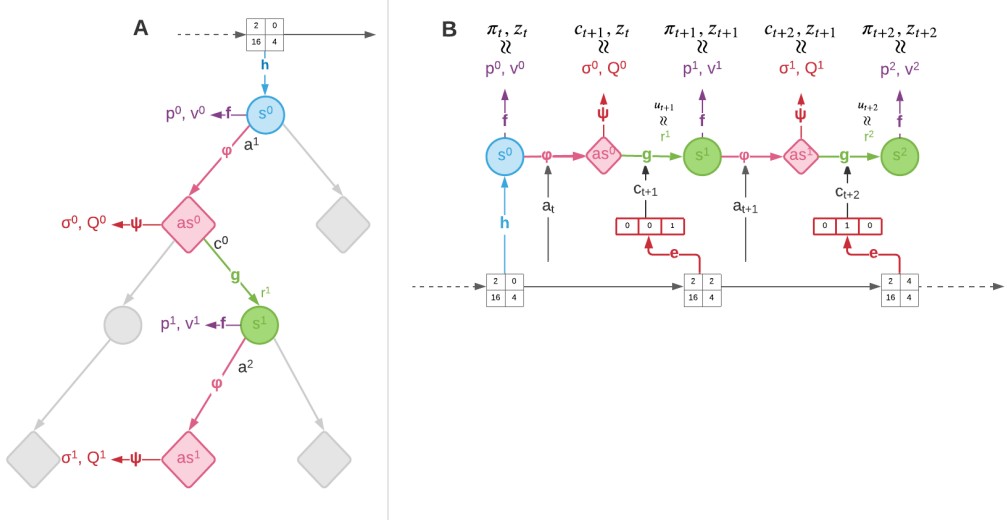

Figure 1: *Stochastic MuZero*. **(A)** Monte Carlo Tree Search used in *Stochastic MuZero*, where diamond nodes represent *chance* nodes and circular nodes represent *decision* nodes. During the *selection* phase edges are selected by applying the pUCT formula in the case of decision nodes, and by sampling the prior $\sigma$ in the case of chance nodes. **(B)** Training of stochastic model in *Stochastic MuZero*. Here for a given trajectory of length 2 with observations $o_{\leq t:t+2}$, actions $a_{t:t+2}$, value targets $z_{t:t+2}$, policy targets $\pi_{t:t+2}$ and rewards $u_{t+1:t+K}$, the model is unrolled for 2 steps. During the unroll, the encoder $e$ receives the observation $o_{\leq t+k}$ as an input and generates a chance code $c_{t+k}$ deterministically. The policy, value and reward outputs of the model are trained towards the targets $\pi_{t+k}$, $z_{t+k}$ and $u_{t+k}$ respectively. The distributions $\sigma^k$ over future codes are trained to predict the code produced by the encoder.

## 4.2 STOCHASTIC SEARCH

*Stochastic MuZero* extends the MCTS algorithm used in *MuZero* by introducing chance nodes and chance values to the search. In the stochastic instantiation of MCTS, there are two types of nodes: *decision* and *chance* (Couetoux, 2013). The *chance* and *decision* nodes are interleaved along the depth of the tree, so that the parent of each *decision* node is a *chance* node. The root node of the tree is always a *decision* node.

In our approach, each *chance* node corresponds to a latent *afterstate* (4.1) and it is expanded by querying the stochastic model, where the parent state and an action are provided as an input and the model returns a value for the node and a prior distribution over future codes $Pr(c \mid as)$. After a *chance* node is expanded its value is backpropagated up the tree. Finally, when the node is traversed during the *selection* phase, a code is selected by sampling the prior distribution [1]. In *Stochastic MuZero* each internal *decision* node is again expanded by querying the learned model, where the state of the *chance* parent node and a sampled code $c$ are provided as an input, and the model returns a reward, a value and a policy. Similarly to *MuZero* the value of the newly added node is backpropagated up the tree, and the pUCT (2) formula is used to select an edge. The stochastic search used by *Stochastic MuZero* is shown schematically in figure 1.

## 5 EXPERIMENTS

We applied our algorithm to a variety of challenging stochastic and deterministic environments. First, we evaluated our approach in the classic game of *2048*, a stochastic single player game. Subsequently, we considered a two player zero-sum stochastic game, Backgammon, which belongs to the same

---

[1]In practice we follow the same quasi-random sampling approach as in Ozair et al. (2021) (A.3), where the code is selected using the formula $\arg\max_c \frac{Pr(c|as)}{N(c)+1}$.

class of board games such as Go, chess or Shogi where *MuZero* excels, but with stochasticity induced by the use of a die. Finally, we evaluated our method in the deterministic game of Go, to measure any performance loss caused by the use of a stochastic model and search in deterministic environments in comparison to *MuZero*.

In each environment we assess our algorithm's ability to learn a transition model and effectively use it during search. To this end, we compare *Stochastic MuZero* (using a stochastic learned model) to *MuZero* (using a deterministic learned model), *AlphaZero* (using a perfect simulator), and a strong baseline method (also using a perfect simulator). In the following sections we present our results for each environment separately.

## 5.1  *2048*

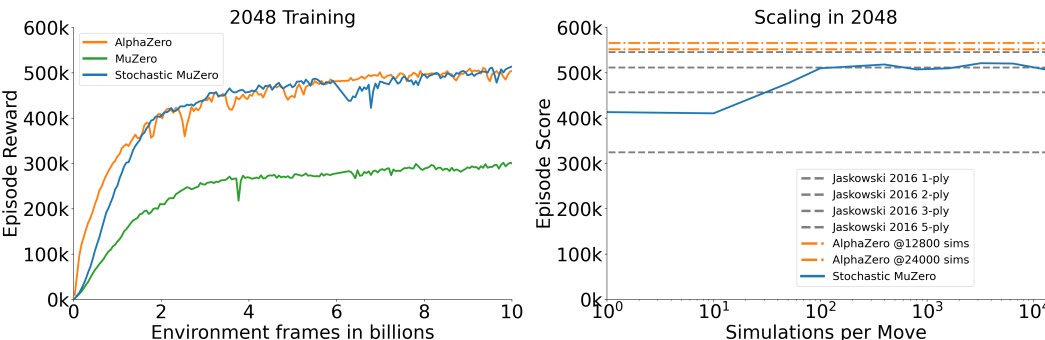

Figure 2: **Planning in *2048***. a) *Stochastic MuZero*, trained using 100 simulations of planning with a learned stochastic model, matched the performance of *AlphaZero*, using 100 simulations of a perfect stochastic simulator, while a deterministic learned model (*MuZero*) performed poorly. b) Evaluation of final agent using different levels of search. *Stochastic MuZero* scales well during evaluation to intermediate levels of search (roughly comparable to 3-ply lookahead), exceeding the playing strength of the state-of-the-art baseline (Jaśkowski, 2016). However, as the number of simulations increases we observe diminishing returns due to imperfections of the learned model.

The game of *2048* (inspired by the game of *Threes!*) is a stochastic, single player, perfect information puzzle game played on a 4x4 board. The objective of the game is to slide numbered tiles on a grid to combine them to create a tile with the number 2048; one can continue to play the game after reaching the goal, creating tiles with larger numbers. The episode reward is the sum of all created tile numbers.

There is a plethora of previous work (Szubert & Jaśkowski, 2014; Yeh et al., 2017; Oka & Matsuzaki, 2016; Rodgers & Levine, 2014; Neller, 2015) on combining reinforcement learning and tree search methods for tackling *2048*. Despite its simplicity, model-free approaches have traditionally struggled to achieve high performance, while planning-based approaches have exploited perfect knowledge of the simulator. To date, the best performing agent used the planning-based approach proposed in (Jaśkowski, 2016). This method used an expectimax tree search over a perfect simulator, combined with domain-specific knowledge and a number of novel algorithmic ideas that exploited the structure of this specific problem.

In contrast our method uses a learned model and no prior knowledge about the environment. Figure 2 compares the performance of *Stochastic MuZero* in *2048* to *AlphaZero*, *MuZero* and the state-of-the-art *Jaskowski 2016* agent. Our method outperformed *Jaskowski 2016*, while using only a quarter of the training data. *Stochastic MuZero* also achieved the same performance as *AlphaZero* (using a perfect simulator), despite learning the model, and performed far better than *MuZero* (using a deterministic model).

## 5.2  BACKGAMMON

Backgammon is a classic two player, zero-sum, stochastic board game; it was popularized as a standard testbed for reinforcement learning and artificial intelligence by TD-gammon (Tesauro, 1995).

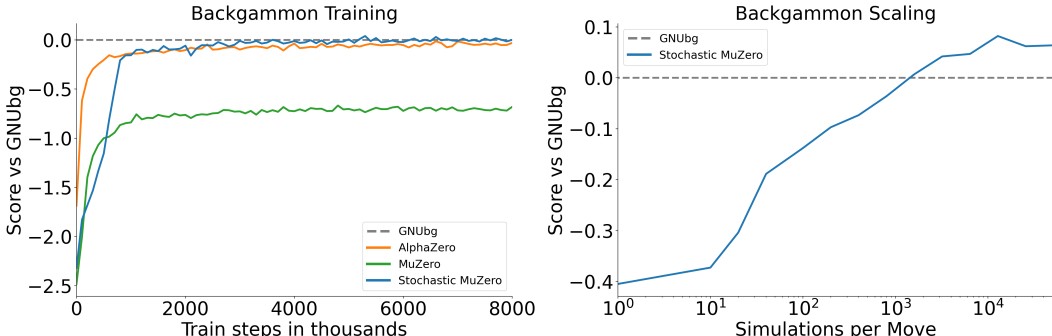

Figure 3: **Stochastic MuZero in Backgammon**. a) *Stochastic MuZero*, trained using 1600 simulations of planning with a learned stochastic model, matched the performance of *AlphaZero*, trained using 1600 simulations of a perfect stochastic simulator, as well as matching the superhuman-level program GNUbg Grandmaster. A deterministic learned model (*MuZero*) performed poorly. b) *Stochastic MuZero*'s model scaled well to large searches, and exceeded the playing strength of GNUbg Grandmaster when using more than $10^3$ simulations.

Here we focus on the single game setting, where the final score takes the values $\pm 1$ for a simple win or loss, $\pm 2$ for a gammon and $\pm 3$ for a backgammon.

In all experiments we compared to GNUbg Grandmaster (Free Software Foundation, 2004), a superhuman-level open-source backgammon player. GNUbg combines a learned value function based on handcrafted features with a specialized min-max tree search using a perfect stochastic simulator. GNUbg Grandmaster uses a 3-ply look-ahead search over a branching factor of 20 legal moves on average and 21 chance transitions.

*Stochastic MuZero*, using a learned stochastic model of the environment and only 1600 simulations per move, achieved the same playing strength as GNUbg, as shown in Figure 5b. The model learned by *Stochastic MuZero* is of high quality: it reached the same playing strength as *AlphaZero* (using a perfect stochastic simulator), and much higher strength than *MuZero* (using a deterministic learned model).

The model also robustly scaled to larger planning budgets (Figure 5c): the performance of *Stochastic MuZero* improved with increasing number of simulations per move, and ultimately exceeded the playing strength of GNUbg Grandmaster.

Given the high dimensionality of the action space in Backgammon (see appendix for details), our Backgammon experiments used the sample-based search introduced by Hubert et al. (2021).

### 5.3 GO

Go is a classic, two player, perfect information, zero-sum board game, that has been studied heavily in the field of artificial intelligence. *AlphaZero* and subsequently, *MuZero* have been the only algorithms which have managed to achieve super-human performance, purely through selfplay, in this challenging domain. Since the goal of *Stochastic MuZero* is to extend the applicability of *MuZero* to stochastic environments while maintaining the latter's performance in deterministic environments, we compared the performance of the two algorithms in the game of Go. Figure 4 shows the Elo (Coulom, 2008) achieved by *Stochastic MuZero* and *MuZero* during training. Although, *Stochastic MuZero* requires twice the number of network expansions in comparison to *MuZero* to achieve the same performance, due to the use of a stochastic MCTS instead of a deterministic one, we ensure that the methods are computationally equivalent by halving the network depth for the chance and dynamic parts of the *Stochastic MuZero*'s network.

### 5.4 REPRODUCIBILITY

In order to evaluate the robustness of our method in all different environments, we replicated our experiments using nine different initial random seeds (see figure 5.4). We observe that our method is

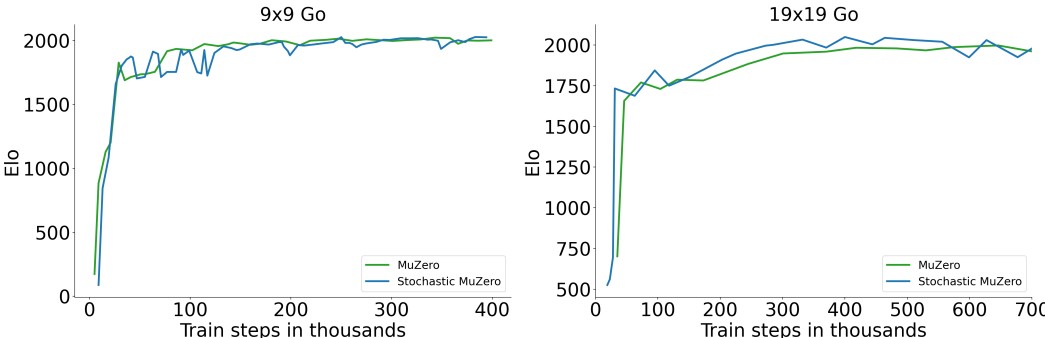

Figure 4: ***Stochastic MuZero* in Go**. Comparison of *Stochastic MuZero* and *MuZero* in the game of Go. a) *Stochastic MuZero* and *MuZero* when compared in 9x9 Go. *MuZero* has a search budget of 200 simulations during training of 800 during evaluation, while *Stochastic MuZero* uses 400 simulations during training and 1600 during evaluation. The Elo scale was anchored so that the performance of the final *MuZero* baseline corresponded to an Elo of 2000. b) *Stochastic MuZero* and *MuZero* when compared in 19x19 Go. *MuZero* has a search budget of 400 simulations during training of 800 during evaluation, while *Stochastic MuZero* uses 800 simulations during training and 1600 during evaluation. The Elo scale was anchored so that the performance of the final *MuZero* baseline corresponded to an Elo of 2000.

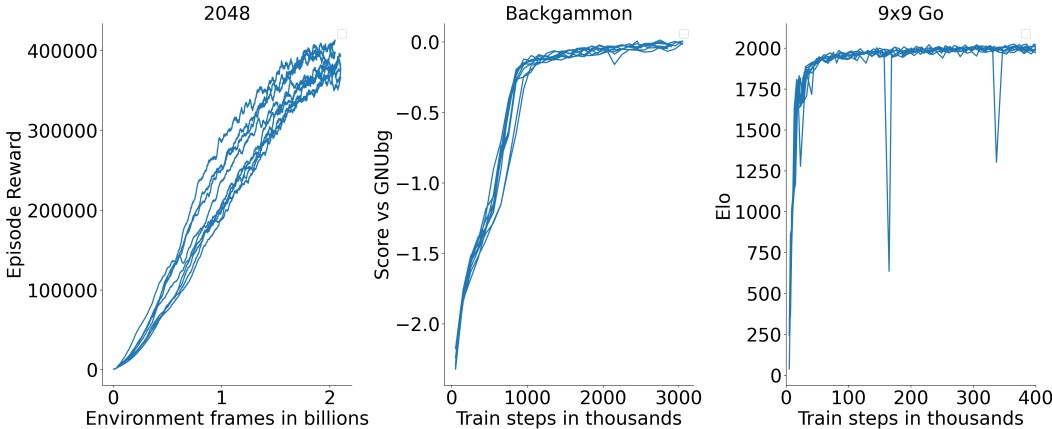

Figure 5: ***Stochastic MuZero* reproducibility across all domains.** We ran our method *Stochastic MuZero* in all environments using 9 different seeds to measure its robustness to random initialization. We observed that there is minimal variation in the performance of *Stochastic MuZero* for all different seeds. Due to the computational cost of each experiment we used a smaller number of training steps for each experiment.

robust to the random initialization and there is minimal variation in its performance between multiple runs. Due to the computational cost of each experiment we used a smaller number of training steps for each experiment.

## 6 CONCLUSIONS

In this work, we proposed a new method for learning a stochastic model of the environment, in a fully online reinforcement learning setting, and showed that the learned model can be effectively combined with planning. Our approach builds on top of *MuZero*, a model-based reinforcement learning agent that has been widely successful in a range of environments and settings, but its applicability is limited to deterministic or weakly stochastic environments. We have shown that our algorithm, *Stochastic MuZero*, can overcome the limitations of *MuZero*, significantly outperforming it in stochastic environments, and it can achieve the same or better performance than *AlphaZero*

which makes use of a perfect simulator for the environment. Finally, we have demonstrated that *Stochastic MuZero* matches or exceeds the performance of previous methods that use a perfect stochastic simulator, in a pure reinforcement learning setting without using any prior knowledge about the environment.

## 7    REPRODUCIBILITY STATEMENT

In order to ensure the reproducability of our results by the research community, we have included detailed pseudocode, references to all environments and datasets used as well as a detailed description of the hyperparameters used (see Appendix). We did not release the full code as it relies on a lot of proprietary internal infrastructure, limiting its usefulness. We also provide a study of the robustness of our method under different random initialization conditions (see 5.4).

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

## A    *Stochastic MuZero* LOSSES

For our Go and Backgammon experiments we used a cross entropy loss for the policy and a MSE loss for the value and reward predictions as shown below:

$$
\begin{aligned}
l^p(\pi, p) &= \pi^T \log p \\
l^v(z, v) &= (z - v)^2 \\
l^r(u, r) &= (u - r)^2
\end{aligned}
\tag{8}
$$

For the extra chance losses used by *Stochastic MuZero*, we used a categorical loss to train the probability distribution over the chance codes and a MSE loss to train the $Q$ afterstate value. The losses are shown bellow:

$$
\begin{aligned}
l^\sigma(c, \sigma) &= \log \sigma(c) \\
l^Q(z, Q) &= (z - Q)^2
\end{aligned}
\tag{9}
$$

In our *2048* experiments, similarly to the *MuZero* Schrittwieser et al. (2020) approach for general MDPs, we used the invertible transform $h(x) = sign(x)(\sqrt{x + 1} - 1 + \epsilon x)$ with $\epsilon = 0.001$ for scaling the value and reward targets and subsequently we obtained a categorical representation for each. Subsequently, we trained the value and reward losses using a categorical loss as shown below:

$$
\begin{aligned}
l^v(z, v) &= Cat(h(z))^T \log \boldsymbol{v} \\
l^r(u, r) &= Cat(h(u))^T \log \boldsymbol{r} \\
l^Q(z, Q) &= Cat(h(z))^T \log \boldsymbol{Q}
\end{aligned}
\tag{10}
$$

## B    *AlphaZero* IMPLEMENTATION

In all our experiments we compared the performance of *Stochastic MuZero* against a custom *AlphaZero* agent. We adapted our *AlphaZero* implementation to achieve good performance in stochastic and single player environments. Specifically:

- We assumed that *AlphaZero* had access to the true distribution of chance outcomes at each transition.
- We used the same conventions of training the policy and value functions of *AlphaZero* as in *MuZero* and *Stochastic MuZero* (i.e. n-step bootstrapping, distributional value representation, value scaling etc.).
- The network architecture for the representation function used by *AlphaZero* was the same as in *MuZero* and *Stochastic MuZero*.
- *AlphaZero* had access to a learned $Q$ value at each chance node, similarly to *Stochastic MuZero*. We found that this significantly improved the performance of the *AlphaZero* agent.

## C    *2048* EXPERIMENTS

The game of 2048 is a single player, perfect information, stochastic puzzle game. The board is represented by a 4x4 grid with numbered tiles, and at each step the player has four possible actions which correspond to the four arrow keys (up, down, right, left). When the player selects an action, all the tiles in the board slide in the corresponding direction until they reach the end of the board or another tile of different value. Tiles of the same value are merged together to form a new tile with a value equal to their sum, and the resulting value is added to the running score of the game. After each move, a new tile randomly appears in an empty spot on the board with a value of 2 or 4. The game ends when there are no more moves available to the player that can alter the board state.

In our *2048* experiments we used a binary representation of the observation as an input to our model. Specifically, the $4 \times 4$ board was flattened into a single vector of size 16, and a binary representation

of 31 bits for each number was obtained, for a total size of 496 numbers. The representation, afterstate dynamics, dynamics and encoder functions were implemented using a 10 block ResNet v2 style pre-activation residual tower (He et al., 2016) coupled with Layer Normalisation (Ba et al., 2016) and Rectified Linear Unit (ReLU) activations. Each linear layer has an output size of 256. We used a codebook of size 32 to model the stochasticity in the environment.

In both the afterstate dynamics and dynamics network the parent states and codes or actions were combined by simple concatenation. The reward and value predictions used the categorical representation introduced in *MuZero* (Schrittwieser et al., 2020). We used 601 bins for both the value and the reward predictions with both the value and the reward being able to represent values between $[0, 600.]$. Furthermore, similarly to *MuZero*, we used the invertible transform $h(x) = sign(x)(\sqrt{x+1} - 1 + \epsilon x)$ with $\epsilon = 0.001$ for scaling the targets. The value targets were computed using $n$-step TD($\lambda$) bootstrapping with $n = 10$, a discount of $0.999$ and $\lambda = 0.5$.

We trained the model using an Adam (Kingma & Ba, 2015) optimizer and a learning rate of $0.0003$ for 20M steps with a batch size of 1024.

We used a prioritized replay buffer, as proposed in (Schaul et al., 2016) and used in the original *MuZero*, with priority $P(i) = \frac{p_i^\alpha}{\sum_k p_k^\alpha}$, where $p_i = \|v_i - z_i\|$, $v$ is the search value and $z$ the observed n-step return, and we set $\alpha$=1. We correct for the sampling bias by scaling the total loss using the importance sampling weights $w_i = (\frac{1}{N} \cdot \frac{1}{P(i)})^\beta$, where we set $\beta$=1. Finally, the replay buffer holds the 125000 most recent sequences, where each sequence has a length of up to 200.

Following *MuZero*, we injected Dirichlet noise to the prior policy at the root node of the search, to allow our agent to explore. We used hyperparameters $alpha = 0.25$ and $fraction = 0.1$ for the injected noise. Furthermore, the agent selected actions by sampling from the visit count distribution at the root node at the end of each search. We used a temperature scheduler with values $[1.0, 0.5, 0.1]$ for the first $[1e5, 2e5, 3e5]$ training steps respectively, and a greedy selection thereafter. We used a budget of 100 simulations for each MCTS search.

## D  BACKGAMMON EXPERIMENTS

Backgammon is an ancient two player, zero-sum, perfect information, stochastic board game. The board consists of 24 squares (or points) and each player controls 15 checkers, which can move based on the outcome of a dice roll. The two players move their checkers in opposite directions and their goal is to move all their checkers off the board first. In addition to a simple winning, a player can also score a double ("gammon") or a triple ("backgammon") winning. A "gammon" is achieved when a player bears off all their checkers before their opponent manages to bear off any, while a "backgammon" when the opponent also has checkers left in the player's home quadrant (farthermost quadrant from the opponent's perspective). Each player can impede the progress of their opponent through "hitting" the opponent's checkers or blocking their advancement. A "hit" is achieved when a player's checker advances to a position with a single opponent's checker. Then the opponent's checker needs to reenter the board in the player's home quadrant and no further moves are allowed to the opponent until that happens. A position is blocked to the opponent when it is occupied by at least two of the player's checkers. Each player makes moves based on the values yielded by rolling two dice. In the case of "doubles", aka the two dice have the same value, the player can play up to 4 moves.

One of the challenges of computer Backgammon is the high branching ratio, since at each ply there are 21 chance outcomes, which yield positions with an average of 20 legal moves each, resulting in a branching ratio of several hundred per ply.

In our backgammon experiments, the board was represented using a vector of size 28, with the first 24 positions representing the number of chips for each player in the 24 possible points on the board, and the last four representing the number of hit chips and born off chips for each of the two players. We used positive numbers for the current player's chips and negative ones for her opponent.

An action in our implementation consists of 4 micro-actions, the same as the maximum number of dice a player can play at each turn. Each micro-action encodes the source position of a chip along with the value of the die used. We consider 26 possible source positions, with the $0^{th}$ position corresponding to a no-op, the $1^{st}$ to retrieving a chip from the hit pile, and the remaining to selecting

a chip in one of the $24$ possible points. Each micro-action is encoded as a single integer with micro-action = src · 6 + die.

Similarly to the *2048* experiments, the representation, afterstate dynamics, dynamics and encoder functions were implemented using a 10 block ResNet v2 style pre-activation residual tower (He et al., 2016) coupled with Layer Normalisation (Ba et al., 2016) and Rectified Linear Unit (ReLU) activations. Each linear layer has an output size of $256$. The action was provided to the afterstate dynamics network as a vector which was the result of the concatenation of the one-hot representation of each micro-action. We used a codebook of size 32 to model the stochasticity in the environment.

Following the work of (Hubert et al., 2021), we used an autoregressive prediction head to model the network policy, with each step corresponding to a single micro-action. To generate a full action, the network was unrolled for $4$ steps. In contrast to the *2048* experiments, the value was represented as a scalar. Similarly to *MuZero* when applied to board games, we used Monte Carlo returns to compute the value targets $z_t$, and we assumed a discount of $1$.

We trained the model using an Adam optimizer with weight decay (Loshchilov & Hutter, 2017), with learning rate of 0.0003 and a weight decay of 0.0001, with a batch size of 1024 for a total of 8M steps.

In all our experiments we used a replay buffer of 100000 games, and the training trajectories were sampled uniformly.

For exploration, we injected dirichlet noise to the prior policy at the root node. However, since the number of legal moves at each position can dramatically change in backgammon, we dynamically adapted the $alpha$ parameter of the dirichlet noise based on the number of legal moves, with $alpha = 1/\sqrt{\text{num\_legal\_moves}}$. We used a budget of 1600 simulations for each MCTS search.

## E  GO EXPERIMENTS

In our Go experiments, we used the same approach as the one proposed in Hubert et al. (2021). The main differences between this setup and the one proposed in the original *MuZero* Schrittwieser et al. (2020) is the use of n-step bootstrapping with a target network to improve the data efficiency of the algorithm.

The *MuZero* and *Stochastic MuZero* players were evaluated during training by playing 100 matches with a search budget of 800 simulations for *MuZero* and 1600 for *Stochastic MuZero*. In order to ensure that the two methods are computationally equivalent, each of the chance and dynamics networks of *Stochastic MuZero* has half the depth of the dynamics network used by *MuZero*. The Elo scale was anchored so that the performance of the final *MuZero* baseline corresponded to an Elo of 2000.

## F  CHANCE ANALYSIS

We investigated the distribution of chance outcomes at each chance node for *Stochastic MuZero*. We collected a dataset for each game by storing the probability distribution over chance nodes, $\sigma_t^k = Pr(c_{t+k+1}|as_t^k)$, for all afterstate prediction network evaluations invoked throughout all searches in 5 episodes. Subsequently, we sorted each chance node distribution and finally, we computed the average distribution, as shown in figure 6 6. We observed that in the case of deterministic environment like Go, the chance distribution collapsed to a single code, while in stochastic environments the model used multiple codes. Furthermore, in Backgammon, the chance distribution had a support of 21 codes with non-negligible probability, which corresponds to the number of distinct rolls of two dice.

## G  COMPUTATIONAL RESOURCES

All experiments were run using second generation Google Cloud TPUs (Google, 2018).

For Backgammon, we used 1 TPU for training and 16 TPUs for acting, for approximately 27 hours - equivalent to 10 days on a single V100 GPU.

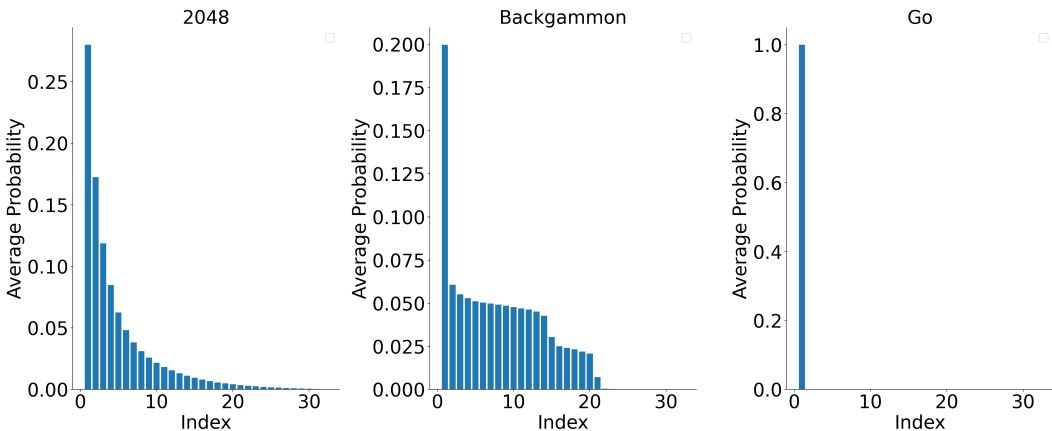

Figure 6: **Average distribution of learned chance outcomes.** The average distribution of learned chance outcomes over all chance nodes after running *Stochastic MuZero* at each game for 5 episodes.

In *2048* we used 1 TPU for training and 4 TPUs for acting, for 80 hours per experiment; equivalent to roughly 8 days on a V100.

Finally, in Go we used the same setup as in *MuZero* (Schrittwieser et al., 2020).

## H IMPLEMENTATION

Stochastic MuZero was implemented as an extension to the standard MuZero algorithm, as it was described in (Schrittwieser et al., 2020). We used the JAX (Bradbury et al., 2018) and Haiku (Hennigan et al., 2020) libraries to implement the neural networks and optimization methods described in this paper. Along with this work, we provide a detailed pseudocode of the *Stochastic MuZero* algorithm, along with all the hyperparameters used for each environment.

## I PSEUDOCODE

For completeness we provide a detailed pseudocode of the *Stochastic MuZero* algorithm, along with all the hyperparameters used by the agent.

```python
"""Pseudocode description of the Stochastic MuZero algorithm.

This pseudocode was adapted from the original MuZero pseudocode.
"""
# pylint: disable=unused-argument
# pylint: disable=missing-docstring
# pylint: disable=g-explicit-length-test
import abc
import math
from typing import Any, Dict, Callable, List, NamedTuple, Tuple, Union,
    Optional, Sequence

import dataclasses
import numpy as np

MAXIMUM_FLOAT_VALUE = float('inf')

########################################
######## Environment interface ##########

# An action to apply to the environment.
# It can a single integer or a list of micro-actions for backgammon.
Action = Any
```

```python
# The current player to play.
Player = int

class Environment:
  """Implements the rules of the environment."""

  def apply(self, action: Action):
    """Applies an action or a chance outcome to the environment."""

  def observation(self):
    """Returns the observation of the environment to feed to the network.
        """

  def is_terminal(self) -> bool:
    """Returns true if the environment is in a terminal state."""
    return False

  def legal_actions(self) -> Sequence[Action]:
    """Returns the legal actions for the current state."""
    return []

  def reward(self, player: Player) -> float:
    """Returns the last reward for the player."""
    return 0.0

  def to_play(self) -> Player:
    """Returns the current player to play."""
    return 0

###########################
####### Helpers ##########

class KnownBounds(NamedTuple):
  min: float
  max: float

class MinMaxStats(object):
  """A class that holds the min-max values of the tree."""

  def __init__(self, known_bounds: Optional[KnownBounds]):
    self.maximum = known_bounds.max if known_bounds else -
        MAXIMUM_FLOAT_VALUE
    self.minimum = known_bounds.min if known_bounds else
        MAXIMUM_FLOAT_VALUE

  def update(self, value: float):
    self.maximum = max(self.maximum, value)
    self.minimum = min(self.minimum, value)

  def normalize(self, value: float) -> float:
    if self.maximum > self.minimum:
      # We normalize only when we have set the maximum and minimum values
          .
      return (value - self.minimum) / (self.maximum - self.minimum)
    return value

# A chance outcome.
Outcome = Any

# An object that holds an action or a chance outcome.
```

```python
ActionOrOutcome = Union[Action, Outcome]

LatentState = List[float]
AfterState = List[float]

class NetworkOutput(NamedTuple):
  value: float
  probabilities: Dict[ActionOrOutcome, float]
  reward: Optional[float] = 0.0

class Network:
  """An instance of the network used by stochastic MuZero."""

  def representation(self, observation) -> LatentState:
    """Representation function maps from observation to latent state."""
    return []

  def predictions(self, state: LatentState) -> NetworkOutput:
    """Returns the network predictions for a latent state."""
    return NetworkOutput(0, {}, 0)

  def afterstate_dynamics(self,
                          state: LatentState,
                          action: Action) -> AfterState:
    """Implements the dynamics from latent state and action to afterstate
        ."""
    return []

  def afterstate_predictions(self, state: AfterState) -> NetworkOutput:
    """Returns the network predictions for an afterstate."""
    # No reward for afterstate transitions.
    return NetworkOutput(0, {})

  def dynamics(self, state: AfterState, action: Outcome) -> LatentState:
    """Implements the dynamics from afterstate and chance outcome to
        state."""
    return []

  def encoder(self, observation) -> Outcome:
    """An encoder maps an observation to an outcome."""

class NetworkCacher:
  """An object to share the network between the self-play and training
      jobs."""

  def __init__(self):
    self._networks = {}

  def save_network(self, step: int, network: Network):
    self._networks[step] = network

  def load_network(self) -> Tuple[int, Network]:
    training_step = max(self._networks.keys())
    return training_step, self._networks[training_step]

# Takes the training step and returns the temperature of the softmax
    policy.
VisitSoftmaxTemperatureFn = Callable[[int], float]

# Returns an instance of the environment.
EnvironmentFactory = Callable[[], Environment]
```

```python
# The factory for the network.
NetworkFactory = Callable[[], Network]

@dataclasses.dataclass
class StochasticMuZeroConfig:
  # A factory for the environment.
  environment_factory: EnvironmentFactory
  network_factory: NetworkFactory

  # Self-Play
  num_actors: int
  visit_softmax_temperature_fn: VisitSoftmaxTemperatureFn
  num_simulations: int
  discount: float

  # Root prior exploration noise.
  root_dirichlet_alpha: float
  root_dirichlet_fraction: float
  root_dirichlet_adaptive: bool

  # UCB formula
  pb_c_base: float = 19652
  pb_c_init: float = 1.25

  # If we already have some information about which values occur in the
  # environment, we can use them to initialize the rescaling.
  # This is not strictly necessary, but establishes identical behaviour
    to
  # AlphaZero in board games.
  known_bounds: Optional[KnownBounds] = None

  # Replay buffer.
  num_trajectories_in_buffer: int = int(1e6)
  batch_size: int = int(128)
  num_unroll_steps: int = 5
  td_steps: int = 6
  td_lambda: float = 1.0
  # Alpha and beta parameters for prioritization.
  # By default they are set to 0 which means uniform sampling.
  priority_alpha: float = 0.0
  priority_beta: float = 0.0

  # Training
  training_steps: int = int(1e6)
  export_network_every: int = int(1e3)
  learning_rate: float = 3e-4
  weight_decay: float = 1e-4

  # The number of chance codes (codebook size).
  # We use a codebook of size 32 for all our experiments.
  codebook_size: int = 32

####################################
## Environment specific configs ##

def twentyfortyeight_config() -> StochasticMuZeroConfig:
  """Returns the config for the game of 2048."""
  def environment_factory():
    # Returns an implementation of 2048.
    return Environment()

  def network_factory():
```

```python
      # 10 layer fully connected Res V2 network with Layer normalization
          and size
      # 256.
      return Network()

  def visit_softmax_temperature(train_steps: int) -> float:
    if train_steps < 1e5:
      return 1.0
    elif train_steps < 2e5:
      return 0.5
    elif train_steps < 3e5:
      return 0.1
    else:
      # Greedy selection.
      return 0.0

  return StochasticMuZeroConfig(
      environment_factory=environment_factory,
      network_factory=network_factory,
      num_actors=1000,
      visit_softmax_temperature=visit_softmax_temperature,
      num_simulations=100,
      discount=0.999,
      root_dirichlet_alpha=0.3,
      root_dirichlet_fraction=0.1,
      root_dirichlet_adaptive=False,
      num_trajectories_in_buffer=int(125e3),
      td_steps=10,
      td_lambda=0.5,
      priority_alpha=1.0,
      priority_beta=1.0,
      training_steps=int(20e6),
      batch_size=1024,
      weight_decay=0.0)

def backgammon_config() -> StochasticMuZeroConfig:
  """Returns the config for the game of 2048."""
  def environment_factory():
    # Returns an backgammon. We consider single games without a doubling
        cube.
    return Environment()

  def network_factory():
    # 10 layer fully connected Res V2 network with Layer normalization
        and size
    # 256.
    return Network()

  def visit_softmax_temperature(train_steps: int) -> float:
    return 1.0

  return StochasticMuZeroConfig(
      environment_factory=environment_factory,
      network_factory=network_factory,
      num_actors=1000,
      visit_softmax_temperature_fn=visit_softmax_temperature,
      num_simulations=1600,
      discount=1.0,
      # Unused, we use adaptive dirichlet for backgammon.
      root_dirichlet_alpha=-1.0,
      root_dirichlet_fraction=0.1,
      root_dirichlet_adaptive=True,
      # Max value is 3 for backgammon.
      known_bounds=KnownBounds(min=-3, max=3),
```

```python
      # 1e5 full episodes stored.
      num_trajectories_in_buffer=int(1e5),
      # We use monte carlo returns.
      td_steps=int(1e3),
      training_steps=int(8e6),
      batch_size=1024,
      learning_rate=3e-4,
      weight_decay=1e-4)

###################################
############ Replay ##############
class SearchStats(NamedTuple):
  search_policy: Dict[Action, int]
  search_value: float

class State(NamedTuple):
  """Data for a single state."""
  observation: List[float]
  reward: float
  discount: float
  player: Player
  action: Action
  search_stats: SearchStats

Trajectory = Sequence[State]

class ReplayBuffer:
  """A replay buffer to hold the experience generated by the selfplay."""

  def __init__(self, config: StochasticMuZeroConfig):
    self.config = config
    self.data = []

  def save(self, seq: Trajectory):
    if len(self.data) > self.config.num_trajectories_in_buffer:
      # Remove the oldest sequence from the buffer.
      self.data.pop(0)
    self.data.append(seq)

  def sample_trajectory(self) -> Trajectory:
    """Samples a trajectory uniformly or using prioritization."""
    return self.data[0]

  def sample_index(self, seq: Trajectory) -> int:
    """Samples an index in the trajectory uniformly or using
       prioritization."""
    return 0

  def sample_element(self) -> Trajectory:
    """Samples a single element from the buffer."""
    # Sample a trajectory.
    trajectory = self.sample_trajectory()
    state_idx = self.sample_index(trajectory)
    limit = max([self.config.num_unroll_steps, self.config.td_steps])
    # Returns a trajectory of experiment.
    return trajectory[state_idx:state_idx + limit]

  def sample(self) -> Sequence[Trajectory]:
    """Samples a training batch."""
    return [self.sample_element() for _ in range(self.config.batch_size)]
```

```python
######################################
############ Search ##############
class ActionOutcomeHistory:
  """Simple history container used inside the search.

  Only used to keep track of the actions and chance outcomes executed.
  """

  def __init__(self,
               player: Player,
               history: Optional[List[ActionOrOutcome]] = None):
    self.initial_player = player
    self.history = list(history or [])

  def clone(self):
    return ActionOutcomeHistory(self.initial_player, self.history)

  def add_action_or_outcome(self, action_or_outcome: ActionOrOutcome):
    self.history.append(action_or_outcome)

  def last_action_or_outcome(self) -> ActionOrOutcome:
    return self.history[-1]

  def to_play(self) -> Player:
    # Returns the next player to play based on the initial player and the
    # history of actions and outcomes. For example for backgammon the two
    # players alternate, while for 2048 it is always the same player.
    return 0

class Node(object):
  """A Node in the MCTS search tree."""

  def __init__(self,
               prior: float,
               is_chance: bool = False):
    self.visit_count = 0
    self.to_play = -1
    self.prior = prior
    self.value_sum = 0
    self.children = {}
    self.state = None
    self.is_chance = is_chance
    self.reward = 0

  def expanded(self) -> bool:
    return len(self.children) > 0

  def value(self) -> float:
    if self.visit_count == 0:
      return 0
    return self.value_sum / self.visit_count

# Core Monte Carlo Tree Search algorithm.
# To decide on an action, we run N simulations, always starting at the
   root of
# the search tree and traversing the tree according to the UCB formula
   until we
# reach a leaf node.
def run_mcts(config: StochasticMuZeroConfig, root: Node,
             action_outcome_history: ActionOutcomeHistory, network:
                 Network,
             min_max_stats: MinMaxStats):
```

```python
  for _ in range(config.num_simulations):
    history = action_outcome_history.clone()
    node = root
    search_path = [node]

    while node.expanded():
      action_or_outcome, node = select_child(config, node, min_max_stats)
      history.add_action(action_or_outcome)
      search_path.append(node)

    # Inside the search tree we use the dynamics function to obtain the
        next
    # hidden state given an action and the previous hidden state.
    parent = search_path[-2]

    if parent.is_chance:
      # The parent is a chance node, afterstate to latent state
          transition.
      # The last action or outcome is a chance outcome.
      child_state = network_output.dynamics(parent.state,
                                            history.
                                                last_action_or_outcome())
      network_output = network_output.predictions(child_state)
      # This child is a decision node.
      is_child_chance = False
    else:
      # The parent is a decision node, latent state to afterstate
          transition.
      # The last action or outcome is an action.
      child_state = network_output.afterstate_dynamics(
          parent.state, history.last_action_or_outcome())
      network_output = network_output.afterstate_predictions(child_state)
      # The child is a chance node.
      is_child_chance = True

    # Expand the node.
    expand_node(node, child_state, network_output, history.to_play(),
                is_child_chance)
    # Backpropagate the value up the tree.
    backpropagate(search_path, network_output.value, history.to_play(),
                  config.discount, min_max_stats)

# Select the child with the highest UCB score.
def select_child(config: StochasticMuZeroConfig, node: Node,
                 min_max_stats: MinMaxStats):
  if node.is_chance:
    # If the node is chance we sample from the prior.
    outcomes, probs = zip(*[(o, n.prob) for o, n in node.children.items()
        ])
    outcome = np.random.choice(outcomes, p=probs)
    return outcome, node.children[outcome]

  # For decision nodes we use the pUCT formula.
  _, action, child = max(
      (ucb_score(config, node, child, min_max_stats), action, child)
      for action, child in node.children.items())
  return action, child

# The score for a node is based on its value, plus an exploration bonus
    based on
# the prior.
def ucb_score(config: StochasticMuZeroConfig, parent: Node, child: Node,
              min_max_stats: MinMaxStats) -> float:
```

```python
    pb_c = math.log((parent.visit_count + config.pb_c_base + 1) /
                     config.pb_c_base) + config.pb_c_init
    pb_c *= math.sqrt(parent.visit_count) / (child.visit_count + 1)

    prior_score = pb_c * child.prior
    if child.visit_count > 0:
      value_score = min_max_stats.normalize(child.reward +
                                            config.discount * child.value()
                                            )
    else:
      value_score = 0
    return prior_score + value_score

# We expand a node using the value, reward and policy prediction obtained
    from
# the neural network.
def expand_node(node: Node, state: Union[LatentState, AfterState],
                network_output: NetworkOutput, player: Player, is_chance:
                    bool):
  node.to_play = player
  node.state = state
  node.is_chance = is_chance
  node.reward = network_output.reward
  for action, prob in network_output.probabilities.items():
    node.children[action] = Node(prob)

# At the end of a simulation, we propagate the evaluation all the way up
    the
# tree to the root.
def backpropagate(search_path: List[Node], value: float, to_play: Player,
                  discount: float, min_max_stats: MinMaxStats):
  for node in reversed(search_path):
    node.value_sum += value if node.to_play == to_play else -value
    node.visit_count += 1
    min_max_stats.update(node.value())
    value = node.reward + discount * value

# At the start of each search, we add dirichlet noise to the prior of the
    root
# to encourage the search to explore new actions.
def add_exploration_noise(config: StochasticMuZeroConfig, node: Node):
  actions = list(node.children.keys())
  dir_alpha = config.root_dirichlet_alpha
  if config.root_dirichlet_adaptive:
    dir_alpha = 1.0 / np.sqrt(len(actions))

  noise = np.random.dirichlet([dir_alpha] * len(actions))
  frac = config.root_exploration_fraction
  for a, n in zip(actions, noise):
    node.children[a].prior = node.children[a].prior * (1 - frac) + n *
        frac

##################################
############ Self-play ###########

class Actor(metaclass=abc.ABCMeta):
  """An actor to interact with the environment."""

  @abc.abstractmethod
  def reset(self):
```

```python
    """Resets the player for a new episode."""

  @abc.abstractmethod
  def select_action(self, env: Environment) -> Action:
    """Selects an action for the current state of the environment."""

  @abc.abstractmethod
  def stats(self) -> SearchStats:
    """Returns the stats for the player after it has selected an action.
      """

class StochasticMuZeroActor(Actor):

  def __init__(self,
               config: StochasticMuZeroConfig,
               cacher: NetworkCacher):
    self.config = config
    self.cacher = cacher
    self.training_step = -1
    self.network = None

  def reset(self):
    # Read a network from the cacher for the new episode.
    self.training_step, self.network = self.cacher.load_network()
    self.root = None

  def _mask_illegal_actions(self,
                            env: Environment,
                            outputs: NetworkOutput) -> NetworkOutput:
    """Masks any actions which are illegal at the root."""
    # We mask out and keep only the legal actions.
    masked_policy = {}
    network_policy = outputs.probabilities
    norm = 0
    for action in env.legal_actions():
      if action in network_policy:
        masked_policy[action] = network_policy[action]
      else:
        masked_policy[action] = 0.0
      norm += masked_policy[action]

    # Renormalize the masked policy.
    masked_policy = {a: v / norm for a, v in masked_policy.items()}
    return NetworkOutput(value=outputs.value, probabilities=masked_policy
        )

  def _select_action(self, root: Node):
    """Selects an action given the root node."""
    # Get the visit count distribution.
    actions, visit_counts = zip(*[
        (action, node.visit_counts)
        for action, node in node.children.items()
    ])

    # Temperature
    temperature = self.config.visit_softmax_temperature_fn(self.
        training_step)

    # Compute the search policy.
    search_policy = [v ** (1. / temperature) for v in visit_counts]
    norm = sum(search_policy)
    search_policy = [v / norm for v in search_policy]
    return np.random.choice(actions, p=search_policy)
```

```python
  def select_action(self, env: Environment) -> Action:
    """Selects an action."""

    # New min max stats for the search tree.
    min_max_stats = MinMaxStats(self.config.known_bounds)

    # At the root of the search tree we use the representation function
        to
    # obtain a hidden state given the current observation.
    root = Node(0)

    # Provide the history of observations to the representation network
        to
    # get the initial latent state.
    latent_state = self.network.representation(env.observation())
    # Compute the predictions.
    outputs = self.network.predictions(latent_state)

    # Keep only the legal actions.
    outputs = self._mask_illegal_actions(env, outputs)

    # Expand the root node.
    expand_node(root, latent_state, outputs, env.to_play(), is_chance=
        False)

    # Backpropagate the value.
    backpropagate([root], outputs.value, env.to_play(),
                  self.config.discount, min_max_stats)
    # We add exploration noise to the root node.
    add_exploration_noise(self.config, root)

    # We then run a Monte Carlo Tree Search using only action sequences
        and the
    # model learned by the network.
    run_mcts(self.config, root, ActionOutcomeHistory(env.to_play()),
             self.network, min_max_stats)

    # Keep track of the root to return the stats.
    self.root = root

    # Return an action.
    return self._select_action(root)

  def stats(self) -> SearchStats:
    """Returns the stats of the latest search."""
    if self.root is None:
      raise ValueError('No search was executed.')
    return SearchStats(
        search_policy={
            action: node.visit_counts
            for action, node in self.root.children.items()
        },
        search_value=self.root.value())

# Self-play.
# Each self-play job is independent of all others; it takes the latest
    network
# snapshot, produces an episode and makes it available to the training
    job by
# writing it to a shared replay buffer.
def run_selfplay(config: StochasticMuZeroConfig,
                 cacher: NetworkCacher,
                 replay_buffer: ReplayBuffer):
  actor = StochasticMuZeroActor(config, cacher)
```

```python
  while True:
    # Create a new instance of the environment.
    env = config.environment_factory()

    # Reset the actor.
    actor.reset()

    episode = []
    while not env.is_terminal():
      action = actor.select_action(env)
      state = State(
          observation=env.observation(),
          reward=env.reward(env.to_play()),
          discount=config.discount,
          player=env.to_play(),
          action=action,
          search_stats=actor.stats())
      episode.append(state)
      env.apply(action)

    # Send the episode to the replay.
    replay_buffer.save(episode)

####################################
############ Training ############

class Learner(metaclass=abc.ABCMeta):
  """An learner to update the network weights based."""

  @abc.abstractmethod
  def learn(self):
    """Single training step of the learner."""

  @abc.abstractmethod
  def export(self) -> Network:
    """Exports the network."""

def policy_loss(predictions, labels):
  """Minimizes the KL-divergence of the predictions and labels."""
  return 0.0

def value_or_reward_loss(prediction, target):
  """Implements the value or reward loss for Stochastic MuZero.

  For backgammon this is implemented as an MSE loss of scalars.

  For 2048, we use the two hot representation proposed in
  MuZero, and this loss is implemented as a KL divergence between the
      value
  and value target representations.

  For 2048 we also apply a hyperbolic transformation to the target (see
      paper
  for more information).

  Args:
    prediction: The reward or value output of the network.
    target: The reward or value target.

  Returns:
    The loss to minimize.
```

```python
    """
    return 0.0

class StochasticMuZeroLearner(Learner):
  """Implements the learning for Stochastic MuZero."""

  def __init__(self,
               config: StochasticMuZeroConfig,
               replay_buffer: ReplayBuffer):
    self.config = config
    self.replay_buffer = replay_buffer
    # Instantiate the network.
    self.network = config.network_factory()

  def transpose_to_time(self, batch):
    """Transposes the data so the leading dimension is time instead of
        batch."""
    return batch

  def learn(self):
    """Applies a single training step."""
    batch = self.replay_buffer.sample()

    # Transpose batch to make time the leading dimension.
    batch = self.transpose_to_time(batch)

    # Compute the initial step loss.
    latent_state = self.network.representation(batch[0].observation)
    predictions = self.network.predictions(latent_state)

    # Computes the td target for the 0th position.
    value_target = compute_td_target(self.config.td_steps,
                                     self.config.td_lambda,
                                     batch)

    # Train the network value towards the td target.
    total_loss = value_or_reward_loss(predictions.value, value_target)

    # Train the network policy towards the MCTS policy.
    total_loss += policy_loss(predictions.probabilities,
                              batch[0].search_stats.search_policy)

    # Unroll the model for k steps.
    for t in range(1, self.config.num_unroll_steps + 1):
      # Condition the afterstate on the previous action.
      afterstate = self.network.afterstate_dynamics(
          latent_state, batch[t - 1].action)
      afterstate_predictions = self.network.afterstate_predictions(
          afterstate)

      # Call the encoder on the next observation.
      # The encoder returns the chance code which is a discrete one hot
          code.
      # The gradients flow to the encoder using a straight through
          estimator.
      chance_code = self.network.encoder(batch[t].observation)

      # The afterstate value is trained towards the previous value target
      # but conditioned on the selected action to obtain a Q-estimate.
      total_loss += value_or_reward_loss(
          afterstate_predictions.value, value_target)

      # The afterstate distribution is trained to predict the chance code
      # generated by the encoder.
```

```python
        total_loss += policy_loss(afterstate_predictions.probabilities,
                                  chance_code)

        # Get the dynamic predictions.
        latent_state = self.network.dynamics(afterstate, chance_code)
        predictions = self.network.predictions(latent_state)

        # Compute the new value target.
        value_target = compute_td_target(self.config.td_steps,
                                         self.config.td_lambda,
                                         batch[t:])
        # The reward loss for the dynamics network.
        total_loss += value_or_reward_loss(predictions.reward, batch[t].
            reward)
        total_loss += value_or_reward_loss(predictions.value, value_target)
        total_loss += policy_loss(predictions.probabilities,
                                  batch[t].search_stats.search_policy)

    minimize_with_adam_and_weight_decay(total_loss,
                                        learning_rate=self.config.
                                            learning_rate,
                                        weight_decay=self.config.
                                            weight_decay)

  def export(self) -> Network:
    return self.network

def train_stochastic_muzero(config: StochasticMuZeroConfig,
                            cacher: NetworkCacher,
                            replay_buffer: ReplayBuffer):
  learner = StochasticMuZeroLearner(config, replay_buffer)

  # Export the network so the actors can start generating experience.
  cacher.save_network(0, learner.export())

  for step in range(config.training_steps):
    # Single learning step.
    learner.learn()
    if step > 0 and step % config.export_network_every == 0:
      cacher.save_network(step, learner.export())

####################################
############ RL loop #############
def launch_stochastic_muzero(config: StochasticMuZeroConfig):
  """Full RL loop for stochastic MuZero."""
  replay_buffer = ReplayBuffer(config)
  cacher = NetworkCacher()

  # Launch a learner job.
  launch_job(lambda: train_stochastic_muzero(config, cacher,
      replay_buffer))

  # Launch the actors.
  for _ in range(config.num_actors):
    launch_job(lambda: run_selfplay(config, cacher, replay_buffer))

# Stubs to make the typechecker happy.
def softmax_sample(distribution, temperature: float):
  return 0, 0

def compute_td_target(td_steps, td_lambda, trajectory):
```

```python
  """Computes the TD lambda targets given a trajectory for the 0th
     element.

  Args:
    td_steps: The number n of the n-step returns.
    td_lambda: The lambda in TD(lambda).
    trajectory: A sequence of states.

  Returns:
    The n-step return.
  """
  return 0.0

def minimize_with_sgd(loss, learning_rate):
  """Minimizes the loss using SGD."""

def minimize_with_adam_and_weight_decay(loss, learning_rate, weight_decay
    ):
  """Minimizes the loss using Adam with weight decay."""

def launch_job(f):
  """Launches a job to run remotely."""
  return f()
```

