# OpenReview forum: "Planning in Stochastic Environments with a Learned Model"
_ICLR.cc/2022/Conference — ICLR 2022 Spotlight_

### Official Review · Reviewer_4qYA · 2021-10-18

**Correctness:** 4
**Technical Novelty And Significance:** 3
**Empirical Novelty And Significance:** 4
**Recommendation:** 10
**Confidence:** 4

**Main Review:**

Strengths:

 - This paper represents a significant contribution to reinforcement learning, by extending the state-of-the-art MuZero to stochastic or partially observed environments.
- The paper is clearly written and clearly explains the methods it builds on.
- The paper is evaluated on both stochastic and non-stochastic environments.

Weakness:
 - I would have liked a discussion of optimality. Expectimax is known to be optimal, but computationally infeasible in many applications. How does this work compare to expectimax and what are the approximations?
 - I would also have liked a discussion of limitations. What are the limitations of the proposed method? For instance, modeling discrete chance outcomes seems to limit this to environments with discrete randomness, e.g. dice rolls, cars, etc. What about environments with continuous stochasticity? Also, how large are the random outcome spaces of the environments tried, and how well does the one-hot chance outcomes scale with larger random outcome spaces? How well does it scale to partially observed environments, like e.g. poker?

**Summary Of The Paper:**

The paper proposes an extension of MuZero to stochastic environments.

The stochasticity of the environment is handled by using afterstates, as_t, and chance outcomes, c_t. This decomposes the modelling of the stochastic environment dynamics into a deterministic model s_{t+1}, r_{t+1} = M(as_t, c_t) and modelling chance outcomes p(c_t | as_t). The chance outcomes are modeled as a discrete categorical variable (1 of M) and learned using a VQ-VAE like setup.

The paper shows how the proposed model achieves ~SOTA on two stochastic environments: 2048 and backgammon, and retains SOTA performance on a single non-stochastic environment, Go, although, using twice the computational budget.

**Summary Of The Review:**

The paper proposes an extension of MuZero to stochastic environments, and achieves ~SOTA results on 2 stochastic environments, 2048 and backgammon, while retaining SOTA performance on a hard non-stochastic environment, Go. The proposed method is a significant step towards a strong, universally applicable, reinforcement learning algorithm.

---

> ### Author Response · Authors · 2021-11-22
> **Reply to reviewer 4qYA**
>
> We would like to thank the reviewer for their thoughtful and insightful review.
>
> > I would have liked a discussion of optimality. Expectimax is known to be optimal, but computationally infeasible in many applications. How does this work compare to expectimax and what are the approximations?
>
> Stochastic MuZero combines a learned, value equivalent model with MCTS search. First, note that, given a perfect model, MCTS converges to the optimal expectimax model, given a sufficient number of simulations (Kocsis and Szepesvari 2007). Second, we have to ask what is meant by a perfect model. Stochastic MuZero aims to learn a value equivalent model, i.e. a model that induces the true value function, for all possible policies that may be encountered during search (Grimm et al 2020). MCTS with respect to a value equivalent model must also converge to the optimal expectimax value. However, given the use of neural networks to  approximate and learn the model, we can offer no guarantee of convergence for the overall algorithm.
> In practice, AlphaZero and MuZero have previously demonstrated that MCTS provides better search outcomes when combined with deep networks in comparison to full-width expectimax search in highly complex environments.
>
> > I would also have liked a discussion of limitations. What are the limitations of the proposed method? For instance, modeling discrete chance outcomes seems to limit this to environments with discrete randomness, e.g. dice rolls, cars, etc. What about environments with continuous stochasticity? Also, how large are the random outcome spaces of the environments tried, and how well does the one-hot chance outcomes scale with larger random outcome spaces? How well does it scale to partially observed environments, like e.g. poker?
>
> We have some preliminary promising results of using Stochastic MuZero in the domain of Leduc Poker, which are not at a state to be presented in this paper. Applying our method in partially observable domains is definitely a really interesting and promising future research direction.

---

### Official Review · Reviewer_TrM2 · 2021-10-22

**Correctness:** 4
**Technical Novelty And Significance:** 3
**Empirical Novelty And Significance:** 4
**Recommendation:** 8
**Confidence:** 4

**Main Review:**

This work can largely be summarized as plugging the ideas from Ozair 2021 into a MuZero training pipeline. Nevertheless, I think it is a valuable contribution to the community, as it addresses a major shortcoming of MuZero and, unlike Ozair 2021, does so in self-play setting without a reconstruction loss.

The paper is nicely written (though see small comments below). My main comments are on the experimental side:

1) Throughout the paper, the submission claims that Stochastic MuZero retains the MuZero’s performance in deterministic settings. However, the paper only demonstrates this in a case in which Stochastic MuZero is given twice the simulation budget of MuZero.  I think that making this claim would require showing that Stochastic MuZero matches the performance of MuZero for equal computation budgets. Maybe Stochastic MuZero’s network architecture could be tuned such that each simulation is only half as costly? Seems like this merits further investigation.

2) One weakness of the paper is that the results on games with stochasticity are not on active (at least in a relative sense) research benchmarks. I think it would be interesting to see Stochastic MuZero benchmarked in Hanabi. While Hanabi is an N-player game, it can easily be turned into a POMDP by fixing N-1 players. This was the approach taken by single-agent SPARTA, for which code has been open sourced: https://github.com/facebookresearch/Hanabi_SPARTA. I think showing Stochastic MuZero’s performance against single-agent SPARTA would be a valuable contribution to the community. This experiment would also move toward addressing reviewer 4qYA's interest (which I also share) in seeing Stochastic MuZero in settings with a larger number of possible outcomes.

3) A third, smaller point, regards the baseline AlphaZero implementation. The appendix gives a number of details about the Stochastic MuZero implementation, but few regarding the AlphaZero implementation. I think it would be beneficial to devote some space to this. For example, did the submission's AlphaZero implementation used Monte Carlo returns, as was done in the AlphaZero paper, or did the submission find it necessary to modify this aspect of AlphaZero to achieve good performance in stochastic settings?

4) A fourth, again smaller point concerns the model learned by Stochastic MuZero. In particular, I would be interested to see some analysis on the extent to which Stochastic MuZero's model performs state abstractions. Ie, does Stochastic MuZero's model look more like the true transition model or more like MuZero's (transition determinization) model? Does Stochastic MuZero end up learning a deterministic model in deterministic environment? How do the shape of Stochastic MuZero's search trees look compared to those of AlphaZero and MuZero?

### Small Comments

It would be more generous to also cite Libratus (not just DeepStack) for tree-based planning algorithms for card games.

“However, learning a model in isolation from its use during planning has proven to be problematic in complex environments”
What about Dreamer v2?

VQ-VAE 3 — make clear this is referring to an equation

code ct+k+1 = one hot (arg maxi (e(o i ≤t+k+1))) produced
The = part of the <= on this line is hard to see because of its proximity to the capital V on the line below.

It took me a minute to figure out how the encoder was being used. Might be worth discussing it at a greater length in the training section.

In Figure 1, it took me a minute to figure out that the pictures of the 2048 game were not pictures of a VQ-VAE embedding. Would be better to use backgammon or something else less misinterpretable.

**Summary Of The Paper:**

The submission proposes an algorithm (called Stochastic MuZero) that combines VQ-VAEs with MuZero. Unlike MuZero, Stochastic MuZero can handle settings with stochasticity in a principled way (in terms of value equivalence). The submission shows that Stochastic MuZero can perform comparably to AlphaZero in 2048 and Backgammon. Additionally, it shows that Stochastic MuZero can perform comparably to MuZero in Go in a setting in which its computation budget is twice as large.

**Summary Of The Review:**

I think that, as a matter of correctness, my 1st point needs to be addressed for the submission to merit acceptance. Simply qualifying the claim would suffice but it would obviously be better if the submission were actually able to substantiate the claim that Stochastic MuZero can match MuZero's performance with an equal computation budget. I also think that the 3rd point should be addressed, toward making the submission's experiments more reproducible. While I think the paper would merit acceptance even without addressing my 2nd and 4th concerns, I will be disappointed if the authors choose not to perform these experiments, as I think they would provide significant value to the community.

---

> ### Author Response · Authors · 2021-11-22
> **Reply to reviewer TrM2**
>
> We would like to thank the reviewer for their insightful comments and suggestions.
>
> > Throughout the paper, the submission claims that Stochastic MuZero retains the MuZero’s performance in deterministic settings. However, the paper only demonstrates this in a case in which Stochastic MuZero is given twice the simulation budget of MuZero. I think that making this claim would require showing that Stochastic MuZero matches the performance of MuZero for equal computation budgets. Maybe Stochastic MuZero’s network architecture could be tuned such that each simulation is only half as costly? Seems like this merits further investigation.
>
> Following the suggestion of the reviewer we reran our Go experiments by halving the number of layers for Stochastic MuZero. This makes the computational cost of the combined dynamics and chance networks comparable to the neural network used in standard MuZero. We observed that our method achieved similar performance to standard MuZero. We have updated our manuscript accordingly.
>
> > One weakness of the paper is that the results on games with stochasticity are not on active (at least in a relative sense) research benchmarks. I think it would be interesting to see Stochastic MuZero benchmarked in Hanabi. While Hanabi is an N-player game, it can easily be turned into a POMDP by fixing N-1 players. This was the approach taken by single-agent SPARTA, for which code has been open sourced: https://github.com/facebookresearch/Hanabi_SPARTA. I think showing Stochastic MuZero’s performance against single-agent SPARTA would be a valuable contribution to the community. This experiment would also move toward addressing reviewer 4qYA's interest (which I also share) in seeing Stochastic MuZero in settings with a larger numbers of possible outcomes.
>
> We respectfully suggest that the notion of "active research benchmarks" may be rather subjective. While agreeing that Hanabi is a great future challenge, note that Go and Backgammon are far more established benchmarks that have received several orders of magnitude more research attention over several decades; and yet no prior program exists that can play both games at superhuman level. Furthermore, Hanabi is largely of interest because it is an imperfect information game. While this may indeed require modeling stochasticity, this also raises other issues, and the reduction of Hanabi to a POMDP against fixed opponents may miss many of the interesting aspects of imperfect information games.
>
> > A third, smaller point, regards the baseline AlphaZero implementation. The appendix gives a number of details about the Stochastic MuZero implementation, but few regarding the AlphaZero implementation. I think it would be beneficial to devote some space to this. For example, did the submission's AlphaZero implementation used Monte Carlo returns, as was done in the AlphaZero paper, or did the submission find it necessary to modify this aspect of AlphaZero to achieve good performance in stochastic settings?
>
> In all experiments we compare AlphaZero and Stochastic MuZero under the same conditions, with two main differences:
> AlphaZero uses the true simulator during search instead of a learned model.
> AlphaZero does not learn a model of the transition dynamics.
> We have added a section in the appendix to clarify this.
>
> > A fourth, again smaller point concerns the model learned by Stochastic MuZero. In particular, I would be interested to see some analysis on the extent to which Stochastic MuZero's model performs state abstractions. Ie, does Stochastic MuZero's model look more like the true transition model or more like MuZero's (transition determinization) model? Does Stochastic MuZero end up learning a deterministic model in deterministic environment? How do the shape of Stochastic MuZero's search trees look compared to those of AlphaZero and MuZero?
>
> Following the reviewer’s suggestion, we have added a new analysis figure to the appendix of the manuscript. This compares the distribution of chance outcomes that is learned by our model, and shows the different “shapes” of distribution that are learned for each domain: collapsing to a single outcome for the deterministic domain of Go, and discovering the 21 possible dice rolls in backgammon.

---

> > ### Comment · Reviewer_TrM2 · 2021-11-22
> > **Follow Up**
> >
> > > Following the suggestion of the reviewer we reran our Go experiments by halving the number of layers for Stochastic MuZero. This makes the computational cost of the combined dynamics and chance networks comparable to the neural network used in standard MuZero. We observed that our method achieved similar performance to standard MuZero. We have updated our manuscript accordingly.
> >
> > Great, thanks for running this experiment.
> >
> > > We respectfully suggest that the notion of "active research benchmarks" may be rather subjective.
> >
> > Fair enough---let me be more concrete. Of the two main experimental domains in the paper, for one, the (as described by the paper) "state of the art" external baseline is from 2016; for the other, there is none given.
> >
> > > While agreeing that Hanabi is a great future challenge, note that Go and Backgammon are far more established benchmarks that have received several orders of magnitude more research attention over several decades; and yet no prior program exists that can play both games at superhuman level.
> >
> > Didn't the submission just show that AlphaZero (a prior program), achieves superhuman level at both? I guess you mean without being given plan-time access to the transition model?
> >
> > > Furthermore, Hanabi is largely of interest because it is an imperfect information game. While this may indeed require modeling stochasticity, this also raises other issues, and the reduction of Hanabi to a POMDP against fixed opponents may miss many of the interesting aspects of imperfect information games.
> >
> > Perhaps I poorly communicated my sentiment above. The fact that imperfect information games raise issues beyond the scope of the submission is immaterial in the context of approaching Hanabi as a POMDP. I am in no way suggesting that reducing Hanabi to a POMDP against fixed opponents retains all of the interesting aspects. The purpose of such experiments would *not* be to solve Hanabi, but rather to better understand Stochastic MuZero. In particular, they would 1) shed light on how Stochastic MuZero handles settings with large numbers of possible outcomes and 2) benchmark Stochastic MuZero against a general purpose, externally benchmarked search algorithm.
> >
> > > Following the reviewer’s suggestion, we have added a new analysis figure to the appendix of the manuscript. This compares the distribution of chance outcomes that is learned by our model, and shows the different “shapes” of distribution that are learned for each domain: collapsing to a single outcome for the deterministic domain of Go, and discovering the 21 possible dice rolls in backgammon.
> >
> > Thanks for running this experiment.
> >
> > -----
> >
> > I raised my score from a 5 to an 8 in response to the revisions.

---

> > > ### Author Response · Authors · 2021-11-23
> > > **Reply to Follow up**
> > >
> > > We would like to thank the reviewer for their constructive reply and comments.
> > >
> > > >Didn't the submission just show that AlphaZero (a prior program), achieves superhuman level at both? I guess you mean without being given plan-time access to the transition model?
> > >
> > > AlphaZero did achieve superhuman level at both, but as the reviewer suggested we meant without access to the true transition model (i.e. without prior knowledge of the rules).
> > >
> > > >Perhaps I poorly communicated my sentiment above. The fact that imperfect information games raise issues beyond the scope of the submission is immaterial in the context of approaching Hanabi as a POMDP. I am in no way suggesting that reducing Hanabi to a POMDP against fixed opponents retains all of the interesting aspects. The purpose of such experiments would not be to solve Hanabi, but rather to better understand Stochastic MuZero. In particular, they would 1) shed light on how Stochastic MuZero handles settings with large numbers of possible outcomes and 2) benchmark Stochastic MuZero against a general purpose, externally benchmarked search algorithm.
> > >
> > > We completely agree that Hanabi is a really interesting and challenging domain and an exciting future research direction for Stochastic MuZero.

---

### Official Review · Reviewer_1oV1 · 2021-11-02

**Correctness:** 3
**Technical Novelty And Significance:** 3
**Empirical Novelty And Significance:** 3
**Recommendation:** 5
**Confidence:** 4

**Main Review:**

I generally liked the ideas in this paper. Though, I believe there is room for improving the related work and empirical results.

## Related  Work
There are two references that were not mentioned as part of the related work, but which are important in the line of value-equivalent MBRL:
* [Model-Based Reinforcement Learning with Value-Targeted Regression](https://arxiv.org/abs/2006.01107)
* [Value Iteration Networks](https://arxiv.org/abs/1602.02867)

## Experiments
To support their claims, the authors chose two environments that exhibit stochasticity: 2048 and Backgammon, and compared the performance of their algorithm against AlphaZero (where a perfect stochastic simulator is used) and MuZero (where a learned deterministic model is used). A similar methodology was used in an additional experiment, using the deterministic, perfect-information game of Go, and here the question was whether their algorithm's learned stochastic model could match the performance of a learned deterministic model.

In general, I see no issue with the choice of domains and baselines. My concerns are mostly with the way results are evaluated and reported. For instance, Figures 2, 3, and 4 all seem to represent a single trial. This is unacceptable for experiments with random outcomes, and it considerably weakens support for the paper's main claim of high performance in stochastic settings. I would imagine the authors expect their results to hold on average, given the randomness of a stochastic model.

Some attempt was made to characterize the variability of results in Section 5.4. Each experiment was repeated---under unknown random conditions---three times for a small fraction of the data shown in figures 2, 3, and 4. There are two issues with this methodology:
1. Three samples is insufficient to characterize a distribution's dispersion.
2. The trials should be run with the same amount of data as the reported results.

To approximate the distribution of performance to a reasonable degree of accuracy, I suggest that a minimum of thirty trials are run for the full length of learning. In addition, the authors should describe exactly the sources of randomness their results represent.

It is also unclear if a hyperparameter search took place for the proposed algorithm.

While the authors do reference relevant work that has used their domains, a concise description of the games and their sources of stochasticity is missing. This is important to communicate explicitly to the reader, since it is the primary environmental feature that experiments depend on. This information could be included in the appendix if the paper is tight on space.

## Minor points
* I didn't find the pseudocode in the appendix very informative. A simple table with all the considered parameters would be sufficient.


**Summary Of The Paper:**

This paper aims to extend previous work on value-equivalent MBRL, such as MuZero, to stochastic environments. In contrast to conventional work in MBRL that fit transition models to be consistent with environmental observations, this line of work fits transition models to improve the accuracy / utility of a downstream value / policy. To this end authors consider the MuZero algorithm and advocate for learning a stochastic model with a VQ-VAE, and they modify MCTS so that it can be used with their stochastic model.

​The authors propose Stochastic MuZero. Their algorithm makes use of a stochastic model to predict future values, policies and rewards. The authors suggest utilizing "afterstates": an imaginary state that is the result of taking an action but it is also before the environment responds with an actual state. In 2048, for example, an afterstate could be the state reached after applying a tile moving action but before a number "2" tile appears in a random place.

As illustrated in Figure 1, the stochastic model consists of 5 functions in contrast to 3 functions in MuZero. The notable addition is in incorporating afterstates in these functions which allows for incorporating chance outcomes. There is an afterstate dynamics function that predicts a latent after-state given a state and action. The typical dynamics function would then still predict a next actual state and reward but its input will be a latent afterstate and a chance outcome. There is also an afterstate prediction function for value and a distribution prediction, where the distribution is that of a chance outcome given an afterstate. That distribution could then be used for sampling chance outcomes in inference.

To adapt MCTS to this model, search starts from a state and then proceeds to alternate at every level between afterstates and states by using the corresponding dynamics function to reach each type of state. ​ ​

**Summary Of The Review:**

Overall, this is an interesting idea, and I encourage the authors to continue working on this paper to provide sound empirical support.  If the claims were sufficiently supported, the results of this paper would be a significant contribution to MBRL. For now, I am leaning towards a rejection as the current empirical methodology is unsound.

### Final
Thank you to the authors, for their response. I have read their replies and the other reviews. I will maintain my original decision to reject, because the biggest issue I have with the paper have not been addressed. This related to the empirical methodology.

---

> ### Author Response · Authors · 2021-11-22
> **Reply to reviewer 1oV1**
>
> We would like to thank the reviewer for their review and suggestions.
>
> > To support their claims, the authors chose two environments that exhibit stochasticity: 2048 and Backgammon, and compared the performance of their algorithm against AlphaZero (where a perfect stochastic simulator is used) and MuZero (where a learned deterministic model is used). A similar methodology was used in an additional experiment, using the deterministic, perfect-information game of Go, and here the question was whether their algorithm's learned stochastic model could match the performance of a learned deterministic model.
> In general, I see no issue with the choice of domains and baselines. My concerns are mostly with the way results are evaluated and reported. For instance, Figures 2, 3, and 4 all seem to represent a single trial. This is unacceptable for experiments with random outcomes, and it considerably weakens support for the paper's main claim of high performance in stochastic settings. I would imagine the authors expect their results to hold on average, given the randomness of a stochastic model.
> Some attempt was made to characterize the variability of results in Section 5.4. Each experiment was repeated---under unknown random conditions---three times for a small fraction of the data shown in figures 2, 3, and 4. There are two issues with this methodology:
> Three samples is insufficient to characterize a distribution's dispersion.
> The trials should be run with the same amount of data as the reported results.
> To approximate the distribution of performance to a reasonable degree of accuracy, I suggest that a minimum of thirty trials are run for the full length of learning. In addition, the authors should describe exactly the sources of randomness their results represent.
>
> We respectfully suggest that our community may be interested in applications of AI close to the forefront of technology, *as well as* understanding their reproducibility. This strongly suggests running singular large experiments with significant computational budgets *as well as* running multiple smaller experiments.
> To further support our claim of the robustness of our method, we have now increased the number of repeated smaller experiments to 9 seeds for each domain. Repeating the long experiments is not computationally feasible. The total number of comparisons available in the paper is now 30 (3*9 short experiments + 3 long experiments), which we believe provides sufficiently strong support for our conclusions.
>
> > It is also unclear if a hyperparameter search took place for the proposed algorithm.
>
> Due to the computational cost of the experiments we did not run a hyperparameter grid search, but instead we manually tuned the different algorithms.
>
> > While the authors do reference relevant work that has used their domains, a concise description of the games and their sources of stochasticity is missing. This is important to communicate explicitly to the reader, since it is the primary environmental feature that experiments depend on. This information could be included in the appendix if the paper is tight on space.
>
> We have added a short description of the sources of stochasticity for each domain in the appendix.

---

> > ### Comment · Reviewer_1oV1 · 2021-11-29
> > **Reply**
> >
> > > We respectfully suggest that our community may be interested in applications of AI close to the forefront of technology, as well as understanding their reproducibility.
> >
> > This is a fine suggestion; the ICLR research community may be interested in AI applications that push the limits of technology. However, this point cannot be used to argue in favor of improper experimental practices or inconclusive results. The fact that a new algorithm is computationally expensive does not permit one to claim statistical significance from a sample size of one or three; it means that the algorithm is inefficient and requires more resources to properly study.
> >
> > I appreciate that the authors repeated the number of small experiments nine times. Those should continue to be repeated so the distribution of each algorithm's performance is characterized with a minimum of thirty seeds. The long experiment should follow the same methodology. Once all experiments have been properly completed, so that the uncertainty of their outcomes has been sufficiently characterized, then I believe this paper would make a significant contribution. Currently, however, it remains incomplete.

---

> > > ### Comment · Area_Chair_WtMw · 2021-11-29
> > > **low numbers of replicates**
> > >
> > > To be clear, there are some statistically significant conclusions that can be drawn from a single sample: e.g., that there is a probability larger than some small threshold of the run completing within the observed time, or that there is probability larger than some small threshold of achieving the observed performance. When compared to methods that verifiably have near-zero probability of doing these things, such conclusions can be useful; it's not always necessary to characterize the exact distribution of uncertainty, and sometimes the expense of doing so can be prohibitive. But care needs to be taken in expressing claims in a situation like this -- it's very easy to overclaim.

---

> > > > ### Comment · Reviewer_1oV1 · 2021-11-29
> > > > **Clarification**
> > > >
> > > > Thank you for this point. I think it's important to narrow the discussion to the context of this paper.
> > > >
> > > > > there is probability larger than some small threshold of achieving the observed performance.
> > > >
> > > > Is this the claim the authors are making? In the context of this paper, this seems equivalent to:
> > > >
> > > > *There exists a seed and a stochastic domain where Stochastic MuZero outperforms MuZero.*
> > > >
> > > > I got the impression the authors wanted to claim something more general; that Stochastic MuZero doesn't just get lucky; it consistently outperforms MuZero with a reasonable level of confidence. My issue is that a reasonable level of confidence has not been established.
> > > >
> > > > If the authors' intention was to show the existence of a good seed, then my recommendation to weakly reject remains the same. In this case, the result would lack significance. How can we be sure this single seed in the chosen domains are reflective of the performance in general stochastic environments?

---

> > > > > ### Author Response · Authors · 2021-11-29
> > > > > **Reply**
> > > > >
> > > > > We appreciate your suggestion. However, 30 repetitions of the long experiment is not computationally feasible. Our conclusion in the paper is that stochastic MuZero significantly outperforms MuZero on stochastic domains, and approximately equals its performance on deterministic games. We have a total of 30 experiments in the paper that all strongly support those statements. Even in the short experiments stochastic MuZero has already outperformed, by a huge margin, the end performance of MuZero in the long Backgammon and 2048 experiments, and already matched the end performance of MuZero in the long Go experiment. As such, we strongly believe that our experiments are fully supportive of our claims.

---

> > > > > ### Comment · Area_Chair_WtMw · 2021-11-29
> > > > > **single seeds**
> > > > >
> > > > > I don't want to put words in the authors' mouths; I'd just like to be fair about what can be claimed from a single seed. We don't want to agree to claims that are too strong; but we also don't want to recommend too far in the other direction. This matters since it is a general issue with relevance to more than just the current submission.
> > > > >
> > > > > In general, from a single seed, we can claim something a lot stronger than just that there exists a seed with the observed properties. Since the seed was randomly chosen (see below for caveats) we can conclude with statistical significance that the observed properties for this seed are in the (for example) [5,95] percentile range containing the bulk of the true distribution. For example, we can reject with good confidence the hypothesis that the algorithm nearly always runs much slower than the observed time.
> > > > >
> > > > > In particular, the statement "the result would lack significance" is not necessarily true; it depends on how the result is phrased. We very much do not want to exclude from publication all experiments where a single run is the most we can afford; such experiments can be highly informative, even in terms of yielding statistically significant conclusions.
> > > > >
> > > > > An example useful conclusion (again, not necessarily what the authors are claiming here) is in the case that there exists a commonly-accepted algorithm with well-characterized performance; if the single seed falls outside the [5,95]% range of this algorithm's known performance, we can conclude that the new algorithm is interesting, in the sense that it likely performs differently -- if in the good direction, it can likely achieve things that the original algorithm could not.
> > > > >
> > > > > Caveats: There are corrections necessary for multiple hypothesis testing (e.g., the authors wouldn't have submitted if their run failed), but these could be managed (e.g., by estimating how many similar papers might have been in development around the submission deadline -- likely at most a handful). Also, since the authors have reported that they could only afford a single run, it makes sense to assume that the single seed was not cherry-picked. Maybe the authors can comment on this; e.g., were there trial runs that were terminated early, or other ways that an additional multiple hypothesis correction might need to be added?
> > > > >
> > > > > Note: this comment isn't intended to cover the shorter runs (that one can afford to repeat).

---

> > > > > > ### Comment · Reviewer_1oV1 · 2021-11-29
> > > > > > **Necessary edits**
> > > > > >
> > > > > > Thank you for continuing the discussion. Your response below was particularly helpful for me, and I believe that it applies well to the current paper's long experiments.
> > > > > >
> > > > > > > An example useful conclusion (again, not necessarily what the authors are claiming here) is in the case that there exists a commonly-accepted algorithm with well-characterized performance; if the single seed falls outside the [5,95]% range of this algorithm's known performance, we can conclude that the new algorithm is interesting, in the sense that it likely performs differently -- if in the good direction, it can likely achieve things that the original algorithm could not.
> > > > > >
> > > > > > I had not considered this sort of claim before. Therefore I am willing to change my score if the authors can agree to two changes.
> > > > > >
> > > > > > 1. Short experiments: Repeat these for 30 trials.
> > > > > > 2. Long experiments: Adjust the writing so the conclusion and claim is softer: it is potentially promising that a single seed of Stochastic MuZero falls outside the established performance range of MuZero in a stochastic domain.

---

> > > > > > ### Author Response · Authors · 2021-12-10
> > > > > > **Reply "single seeds"**
> > > > > >
> > > > > > >Caveats: There are corrections necessary for multiple hypothesis testing (e.g., the authors wouldn't have submitted if their run failed), but these could be managed (e.g., by estimating how many similar papers might have been in development around the submission deadline -- likely at most a handful). Also, since the authors have reported that they could only afford a single run, it makes sense to assume that the single seed was not cherry-picked. Maybe the authors can comment on this; e.g., were there trial runs that were terminated early, or other ways that an additional multiple hypothesis correction might need to be added?
> > > > > >
> > > > > >
> > > > > > In all our experiments we chose the hyperparameters based on short experiments similar to the ones shown in the reproducibility section. The longer runs were obtained by running the same experiment longer *once* without any additional corrections (i.e. early termination of underperforming experiments).

---

### Official Review · Reviewer_qs8f · 2021-11-03

**Correctness:** 4
**Technical Novelty And Significance:** 3
**Empirical Novelty And Significance:** 4
**Recommendation:** 8
**Confidence:** 4

**Main Review:**

Potential Areas of Improvements / Questions:
- One thing I would have liked to see, possibly in the appendix, is confirmation that this model trained on Go-Zero learned to become fully, or almost-fully deterministic. It’s not essential since it clearly works, but I think it’d be interesting! I suspect it doesn’t completely do this as this is the only explanation I am able to come up for why you need a larger sampling budget for stochastic mu-zero?
- To the former point, why do you need a larger sampling budget for stochastic mu-zero in the Go games?
- Would this model also then be applicable to multi-player, imperfect information games? This also does not affect my score nor am I asking for additional experiments on this, I just think such an experiment would be interesting.
- Is the notion of after-states strictly needed? Why is it advantageous to explicitly split the deterministic and stochastic components instead of having each node be stochastic?
- It would be good to include the hyperparam grids that were searched over to get the results in the appendix, or if none was done, to say so. The robustness section is nice but it’s valuable to know how much tuning was involved.

Clarity:
- In the “Model” section, I might have missed it, but it looks like l^p is not defined? Same for l^v.
- Notational question: why is c_{t+k+1} the only thing where the k index is a subscript rather than a superscript?
- In the section “chance outcomes” there seems to be something wrong with the sentence “By using a fixed codebook of one hot vectors, we can simplify the equations of the VQ-VAE 3”
- I may have missed it but in case I didn’t, the appendix should include an exact description of the training details of VQ-VAE, including when different components are frozen, the size of the codebook, etc. Unless I missed something, the paper is not reproducible as is.
- After equation (5), I am not totally clear what is meant by the expression $\sigma_t^k, ,c_{t+k+1}\sim \sigma_t^k$. I assume you mean to say that chance outcomes are drawn from sigma but the construction of the sentence makes it read like you are drawing $\sigma_t^k$ from $\sigma_t^k$.
- I think it would be worthwhile, at least in the appendix, to be clearer about the process by which the chance variables are actually sampled. The description in the “chance outcomes” section “The resulting encoder can also be viewed as a stochastic it
function of the observation which makes use of the Gumbel softmax reparameterization trick (Jang et al., 2016) with zero temperature during the forward pass and a straight through estimator during the backward”  is a little unclear.
- Given that the VQ-VAE model is a key contribution, I really do think clearer explanation of how the “novel” variant of VQ-VAE works could be given. I don’t have a substantive solution, I just want the authors to be aware that this section was slightly confusing to read. In particular, it might be worth expanding on how the deterministic codes let you still acquire stochasticity.
- Is $l^\sigma$ defined anywhere?
- I think in Fig. 1B you want the sampling symbol rather than the approximately equal symbol?

**Summary Of The Paper:**

The authors extend mu-zero to MDPs with stochasticity by adding after-states to the tree and using a VQ-VAE model. They show that this enables them to solve stochastic environments where mu-zero fails.

**Summary Of The Review:**

A very good paper with a couple of areas that need some work in rewriting to make the method clearer.

---

> ### Author Response · Authors · 2021-11-22
> **Reply to reviewer qs8f**
>
> We would like to thank the reviewer for their thoughtful and insightful review.
>
> > One thing I would have liked to see, possibly in the appendix, is confirmation that this model trained on Go-Zero learned to become fully, or almost-fully deterministic.
>
> The resulting stochastic model is deterministic and we can measure this by observing that the prior distribution $\sigma(c | as)$ collapses over a single code for each transition. We have added a section in the appendix (Chance Analysis) to show this.
>
> > To the former point, why do you need a larger sampling budget for stochastic mu-zero in the Go games?
>
> In the game of Go, searching deeper for more plies leads to a substantially higher performance. In Stochastic MuZero each ply corresponds to 2 network evaluations: one for the expansion of the afterstate, and one for the actual state. As a result, in order to achieve a comparable depth of plies Stochastic MuZero requires more network evaluations. We had previously observed, experimentally, that by allowing Stochastic MuZero to use twice the number of network evaluations leads to a comparable performance. To make this point clearer, we have now updated our experiments to use equivalent computational cost, by halving the depth of the chance and dynamics networks, in comparison to standard MuZero. Our results showed the same overall performance as MuZero, even when using a comparable computational cost per search.
>
> > Would this model also then be applicable to multi-player, imperfect information games?
>
> We believe that Stochastic MuZero could be applied to multi-player of imperfect information games and this would indeed be a really interesting future research direction.
>
> > Is the notion of after-states strictly needed? Why is it advantageous to explicitly split the deterministic and stochastic components instead of having each node be stochastic?
>
> Searching effectively over a stochastic game tree involves dealing with the depth-breadth trade-off when constructing the tree. In our work we found that by explicitly modeling the transition probabilities through the use of afterstates and a prior over real states provided us with two main benefits:
> Training the model in an end-to-end fashion proved to be more robust and accurate.
> When searching with a stochastic learned model which produces state samples it is not straightforward to cluster samples which correspond to the same underlying state correctly (we would need some distance function with an epsilon hyperparameter). Our discrete model achieves exactly that during training in an end-to-end fashion.
> When traversing a chance node we sample from the true transition probability instead of a biased distribution we could obtain by using progressive widening or some other approach.
>
> > It would be good to include the hyperparam grids that were searched over to get the results in the appendix, or if none was done, to say so.
>
> Due to the computational cost of the experiments we did not run a hyperparam grid search, but instead we manually tuned the different algorithms. We observed that our method is sufficiently robust to different hyperparameters.
>
> > In the “Model” section, I might have missed it, but it looks like l^p is not defined? Same for l^v.
>
> In this section, we use the same notation as the one used in the MuZero paper. We have updated the manuscript to clarify the meaning of these losses in the appendix.
>
> > Notational question: why is c_{t+k+1} the only thing where the k index is a subscript rather than a superscript?
>
> We use subscripts to notate real steps and superscripts to notate model steps. Since during training the code is produced by the posterior which has access to the real observation at time t+k+1 we use a subscript in this notation.
>
> > In the section “chance outcomes” there seems to be something wrong with the sentence “By using a fixed codebook of one hot vectors, we can simplify the equations of the VQ-VAE 3”
>
> VQ-VAE uses a learned codebook of embeddings. In all our experiments we use a codebook of one hot vectors which does not change during training (fixed).
>
> > I may have missed it but in case I didn’t, the appendix should include an exact description of the training details of VQ-VAE, including when different components are frozen, the size of the codebook, etc.
>
> Differently to the original VQ-VAE, our approach is trained end-to-end with no separate step for learning the codebook. The codebook is always fixed (aka not learned) and corresponds to a set of one-hot vectors. In the appendix for each experiment and in the pseudocode we define the codebook size to be 32.
>
> > After equation (5), I am not totally clear what is meant by the expression σtk,,ct+k+1∼σtk.
>
> We would like to thank the reviewer for their comment, we have rephrased this in our updated version.
>
> > I think in Fig. 1B you want the sampling symbol rather than the approximately equal symbol?
>
> The almost equal sign is used to clarify what the target for each output is.

---

### Decision · Program_Chairs · 2022-01-20

**Decision:**

Accept (Spotlight)

**Comment:**

The paper extends MuZero to stochastic (but observable) MDPs. To represent stochastic dynamics, it splits transitions into two parts: a deterministic transition to an afterstate (incorporating all observations and actions up to the current time), followed by a stochastic outcome (accounting for new randomness that follows the last action). The transition to an afterstate is similar in spirit to ordinary MuZero's dynamics model; the stochastic outcome is learned by a VQ-VAE. At planning time, MuZero retains the MCTS lookahead from ordinary MuZero. Stochastic MuZero achieves impressive results: e.g., it maintains the original MuZero's strong performance on the deterministic game of Go, while improving on MuZero significantly (and achieving superhuman performance) on the stochastic game of backgammon.

This is a strong paper overall: it presents a convincing and successful extension of the already-influential MuZero work, along with large-scale computational experiments confirming the utility of the approach. There are nonetheless a few weaknesses: first, compared to the original AlphaZero and MuZero work, it is perhaps less surprising that the given approach is successful, since it is more closely related to prior work. Second, due to the large-scale computational infrastructure needed, it is only possible to run some of the experiments once. This is not in itself a problem, but care needs to be taken in interpreting the results of such single-run experiments: e.g., any figures that show results of single-run experiments should have a clear warning label, and any statements such as "stochastic MuZero performs better than original MuZero" should be tempered with a caveat about how reliable these conclusions are likely to be. Section 5.4 (which runs shorter experiments using three random seeds each) makes a start at evaluating reliability, but (a) the headline results in previous sections do not contain any caveats or pointers to 5.4, and (b) 5.4 should explicitly acknowledge that it cannot hope to detect even quite-common failure cases with so few seeds.